# PROCESSLID: STEP-WISE INTERNAL REWARD IN LLM REASONING VIA LOCAL INTRINSIC DIMENSION

## ABSTRACT

Accurate step-level correctness signals are key to reliable LLM mathematical reasoning. Prior work either invokes external judges or Process Reward Models at inference time, incurring heavy compute, or leverages internal representations but yields outcome-level signals without step-wise granularity. We introduce **ProcessLID**, a training-free, representation-based method grounded in local intrinsic dimension that produces step-level correctness signals. Across six models on four math benchmarks, ProcessLID attains the state-of-the-art step-level and outcome-level performance and remains competitive in inference-time settings. Lastly, we provide analyses explaining the effectiveness of our methodology.

## 1 INTRODUCTION

LLM mathematical reasoning underpins applications from education to scientific discovery. Recent work shows that enabling multi-step reasoning is central to solving challenging math problems (Wei et al., 2023; DeepSeek-AI et al., 2025). However, LLMs remain prone to logical errors within long Chain of Thought (CoT), a trajectory of multiple reasoning steps (Lightman et al., 2023).

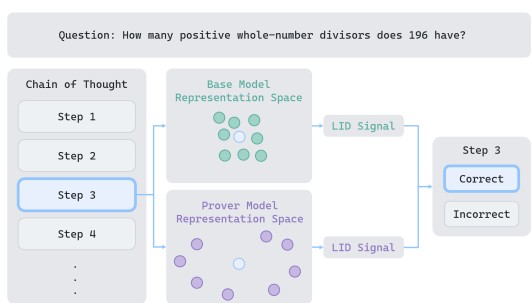

Figure 1: Illustration of ProcessLID

To mitigate this issue, a prominent approach trains Process Reward Models (PRMs) on large collections of labeled steps to assign scalar rewards that reflect the validity of a reasoning step (Lightman et al., 2023). PRMs provide reliable step-level correctness signals, but this approach has two limitations. First, training strong PRMs and obtaining step-level labels through human annotation or Monte Carlo estimation is cost- and compute-intensive (Wang et al., 2024). Second, PRMs can struggle to generalize even within the domain they were trained on when task difficulty shifts (Zhu et al., 2025).

Motivated by these costs and robustness concerns, internal signals from LLMs have been investigated to estimate reasoning step correctness produced by the LLM itself. One line uses logit distributions to form uncertainty-based signals such as entropy or confidence (Hendrycks & Gimpel, 2016; Guo et al., 2017; Xie et al., 2024; Huang et al., 2025). These distributions often quantify uncertainty or consistency rather than correctness and can be miscalibrated under out-of-distribution inputs (Desai & Durrett, 2020; Farquhar et al., 2024). Another line aggregates embeddings produced by LLMs over an entire CoT to form outcome-level scores (Chen et al., 2024; Grewal et al., 2024). However, representation-based approaches rarely produce signals at step-level granularity, and some rely on representation geometry heuristics that do not transfer consistently across models.

We ask a focused question: can we obtain a training-free, step-level correctness signal?

In this paper, we present **ProcessLID**, an internal representation based and training-free correctness signal that operates at step-level[1] granularity. The method is grounded on the concept of Local Intrinsic Dimension (LID), which quantifies the complexity of the distribution that a representation

---

[1] We use "step" and "process" interchangeably; both refer to the an intermediate reasoning step within a CoT.

lies in. Prior work in short Question-Answering (QA) settings shows that embeddings of correct answers generated by LLMs tend to have lower LID values than incorrect ones (Yin et al., 2024). This is because incorrect answers are sampled from fabricated model distribution, which is typically more complicated than natural language distribution, and thus has higher intrinsic dimensions. Building upon similar intuition, we hypothesize that LID is also informative for the correctness of CoTs in mathematical reasoning: along a correct prefix[2], each step introduces constraints (e.g., fixing a method or simplifying expressions) that narrow the remaining solution space, compressing the reachable representation states near the prefix and yielding lower LID; incorrect prefixes introduce contradictions or irrelevant branches that expand plausible continuations, leading to higher LID.

Building on this intuition, we introduce two innovations to make LID effective at the step level: (i) we compare LID of the CoT before and after adding a step to isolate that step's correctness, and (ii) we compute a complementary LID signal using a prover LLM that is distinct from the generator. Across six models and four math benchmarks, ProcessLID achieves an average improvement of 7.33% under AUROC over strong baselines and remains competitive in inference time settings.

Our contributions can be summarized as follows:

1. We propose ProcessLID, a training-free, step-level correctness signal based on LID.

2. We show that our method demonstrates state-of-the-art performance across six models and four math benchmarks in both step-level and outcome-level settings.

3. We show that our method is competitive with a strong reward model in inference time settings.

4. We provide interpretability analyses that explain the effectiveness of our design choices.

## 2 RELATED WORK

**Judgment-based evaluators and self-assessment** This approach utilizes explicit judgments either from dedicated evaluators (judge LLMs, PRMs) or from the generator's own self-assessment to score step correctness. Judge LLMs and PRMs characterize step-level correctness via verbal decisions or scalar rewards, but require substantial compute to sample full CoT or to train the evaluator (Ye et al., 2025a; Zheng et al., 2023; Gu et al., 2024). Generator-driven variants prompt the LLM itself to verbalize its confidence through additional generations, avoiding a separate evaluator but still incur extra sampling and exhibit brittleness to prompting/calibration choices (Kadavath et al., 2022).

**Representation-driven signals** This approach derives correctness signals directly from a model's internal representation. At the token level, uncertainty signals such as entropy and maximum softmax probability are computed from the predictive probabilities generated by LLMs; these signals can be aggregated into outcome-level signals and have been adapted to reasoning-centric settings (Guo et al., 2017; Xie et al., 2024; Huang et al., 2025). At the outcome level, embeddings across tokens over the CoT are first aggregated into a single embedding; correctness signals are then derived either by extracting correlated features from this aggregate representation or by decomposing it spectrally or semantically to detect hallucinations (Chen et al., 2024; Grewal et al., 2024). While they avoid the heavy generation/training of judgment-based approaches, these methods predominantly yield token-level or outcome-level signals and seldom align with the desired step-level granularity.

## 3 PROBLEM SETUP

Let $\pi$ be an $L$-layer causal LLM. Given a question $Q$, $\pi$ produces a reasoning trajectory $S = [s_1, s_2, \ldots, s_H]$, where $H$ is the number of reasoning steps and $s_j$ is the $j$-th step. We write $S_{1:j}$ for the prefix containing the first $j$ steps.

Let $\mathcal{D} = \{Q^{(1)}, \ldots, Q^{(n)}\}$ be a set of questions. For each $Q^{(i)}$, the model $\pi$ produces a trajectory $S^{(i)} = [s_{i,1}, \ldots, s_{i,H_i}]$ with step-level ground-truth labels $\hat{Y}^{(i)} = \{\hat{y}_{i,1}, \ldots, \hat{y}_{i,H_i} : \hat{y}_{i,j} \in \{0,1\}\}$ where $\hat{y}_{i,j} = 1$ indicates the step $s_{i,j}$ is correct. These labels can be obtained from human annotators or estimated via additional rollouts from $\pi$ or external judge LLMs (Wang et al., 2024).

---

[2]We use prefix to denote a partial CoT consisting of the first $j$ steps of a trajectory.

We introduce a scoring function $g : (Q^{(i)}, S_{1:j}^{(i)}) \to \mathbb{R}$ which assigns a correctness score to step $s_{i,j}$ given its prefix $(Q^{(i)}, S_{1:j}^{(i)})$; larger values indicate a higher likelihood that $s_{i,j}$ is correct. Our goal in step-level correctness characterization is for $g(Q^{(i)}, S_{1:j}^{(i)})$ to achieve high classification accuracy against the ground-truth $\hat{y}_{i,j}$ for all steps $j \in \{1, \ldots, H_i\}$ across all trajectories $\{S^{(1)}, \ldots, S^{(n)}\}$. For outcome-level, the objective is for $g(Q^{(i)}, S^{(i)})$ (the score of the final step $s_{i,H_i}$ in each trajectory $S^{(i)}$) to achieve high classification accuracy against the ground-truth across all questions in $\mathcal{D}$.

## 4 METHODOLOGY

### 4.1 LOCAL INTRINSIC DIMENSION (LID)

Given a question $Q^{(i)} \in \mathcal{D}$, consider its $j$-th reasoning step $s_{i,j}$ and representation $E_{\pi,j}^{(i)} \in \mathbb{R}^D$ generated by $\pi$. Let $\{E_{\pi,j}^{(t)}\}_{t=1}^T$ denote the $T$ nearest neighbors of $E_{\pi,j}^{(i)}$, where each neighbor is the representation at the $j$-th reasoning step drawn from CoTs for other questions in the same dataset $\mathcal{D}$. The LID of $E_{\pi,j}^{(i)}$ is

$$m_\pi\big(E_{\pi,j}^{(i)}\big) \;=\; \left( \frac{1}{T-1} \sum_{k=1}^{T-1} \frac{d_T}{d_k} \right)^{-1} ,$$

where $d_k$ is the Euclidean distance from $E_{\pi,j}^{(i)}$ to its $k$-th nearest neighbor in $\{E_{\pi,j}^{(t)}\}_{t=1}^T$.

Yin et al. (2024) introduce LID as an outcome-level correctness signal in short, single-turn QA, observing that representations of incorrect answers lie in more complicated distributions than correct ones. We adopt LID as a foundational primitive and hypothesize it is also informative for CoTs in mathematical reasoning: along a correct prefix, each step adds constraints (e.g., fix a method, simplify expressions) that narrow the remaining solution space, compressing the reachable representation states around the prefix and yielding lower LID; incorrect prefixes tend to introduce contradictions or irrelevant branches, expanding plausible continuations and producing higher LID.

### 4.2 MEASURING STEP-LEVEL CORRECTNESS VIA STEP LID DELTA

Directly using $m_\pi(E_{\pi,j})$ as a step-level signal can be misleading. In decoder-only Transformers (Vaswani et al., 2023), the representation for step $s_j$ is contextualized by the entire prefix $S_{1:j}$. Thus $E_{\pi,j}$ (and its LID) reflects both the current step $s_j$ and all preceding steps, confounding the contribution of $s_j$. To isolate a step's marginal correctness, we compare the LID of the trajectory before and after adding $s_j$ and define the **Step LID Delta**: $\Delta m_\pi\big(E_{\pi,j}\big) := m_\pi\big(E_{\pi,j}\big) - m_\pi\big(E_{\pi,j-1}\big)$.

This difference quantifies how much $s_j$ changes the LID relative to its prefix. Small increments indicate a move toward a more structured manifold, suggesting $s_j$ improves correctness. Larg incre-ments indicate the manifold becomes more heterogeneous, suggesting potential error in $s_j$.

### 4.3 UTILIZING DISTINCT REPRESENTATION GEOMETRY VIA A PROVER MODEL

Setlur et al. (2024) finds that, in reinforcement learning, estimating step-level rewards with a prover policy different from the base policy can substantially improve the base policy's performance. They argue that the prover provides additional variance to discriminate the quality of the base policy's actions while remaining aligned with the task objective. Motivated by this advantage, we introduce a prover model $\mu$ to measure step-level correctness under a distinct representation geometry.

Concretely, let $\mu$ be the prover LLM, not necessarily equal to $\pi$. For each reasoning step $s_j$ generated by $\pi$, we feed the prefix $S_{1:j}$ to $\mu$ and extract the prover representation $E_{\mu,j}$ at the last token of $s_j$. We then define the **LID Embedding Delta** under the prover: $m_\mu\big(\Delta E_{\mu,j}\big) := m_\mu\big(E_{\mu,j} - E_{\mu,j-1}\big)$.

When computing $m_\mu(\Delta E_{\mu,j})$, the neighbor set is formed from the deltas of embeddings $\{\Delta E_{\mu,j}^{(t)}\}_{t=1}^T$ rather than the raw embeddings $\{E_{\mu,j}^{(t)}\}_{t=1}^T$. Both $\Delta m_\pi(E_{\pi,j})$ and $m_\mu(\Delta E_{\mu,j})$ aim to measure the genuine correctness of the current step from complementary perspectives. The former compares LID before and after adding $s_j$ in the base model geometry, while the latter computes the

LID of the embedding change in the prover geometry, focusing on the new information injected by $s_j$. Computing $E_{\mu,j}$ requires only a single forward pass through $\mu$; no rollouts from $\mu$ are required.

## 4.4 PROCESSLID

We propose PROCESSLID, a simple additive combination of the Step LID Delta and the LID Embedding Delta:

$$R(E_{\pi,j}) = \Delta m_\pi(E_{\pi,j}) + \alpha\, m_\mu(\Delta E_{\mu,j}), \alpha \in \mathbb{R}_{>0}.$$

where $\alpha$ is a hyperparameter. We hypothesize the prover term $m_\mu(\Delta E_{\mu,j})$ enhances discriminative power by contributing a signal that is partly orthogonal to the base-model signal $\Delta m_\pi(E_{\pi,j})$ while retaining sufficient variance to separate truthful from untruthful steps generated by $\pi$. The asymmetric formulation is also intentional: measuring the base model with a Step LID Delta and the prover with an LID Embedding Delta maximizes the amount of new information in the prover's signal that is not captured by the base model, compared with symmetric alternatives that use both Step LID Deltas $\left(\Delta m_\pi(E_{\pi,j}) + \alpha\,\Delta m_\mu(E_{\mu,j})\right)$ or both Embedding Deltas $\left(m_\pi(\Delta E_{\pi,j}) + \alpha\,m_\mu(\Delta E_{\mu,j})\right)$.

## 4.5 AGGREGATING PROCESSLID ACROSS LAYERS

LLMs produce a representation at each layer for every token. Thus, for an L-layer LLM $\pi$ and a reasoning step $s_j$, we obtain $L$ per-layer ProcessLID signals $R(E_{\pi,j,1}), \ldots, R(E_{\pi,j,L})$. Given that different layers have different discriminative power, prior work (Yin et al., 2024) selects a single layer via heuristics. We propose a more generalizable approach that aggregates signals over a subset of layers to account for the fact that multiple layers can carry strong signals. We obtain this subset $\mathcal{L}^* \subseteq \{1, \ldots, L\}$ by selecting the layers that perform best on a disjoint calibration set.

Subsequently, we convert each per-layer signal $R(E_{\pi,j,l})$ into a layer-wise $p$-value $p_{\pi,j,l}$ using the same calibration set, then aggregate over the selected layers with Fisher's method (Fisher, 1970): $T_j = -2 \sum_{l \in \mathcal{L}^*} \ln\left(p_{\pi,j,l}\right)$.

Larger $T_j$ indicates the step $s_j$ is more likely incorrect. Implementation details for $p$-value normalization and the procedure to obtain $\mathcal{L}^*$ are provided in Appendix A.2.

## 5 EXPERIMENT

We compare ProcessLID to other methods with two tasks:

1. Offline correctness classification: we aim to verify if ProcessLID could help detect incorrect CoT reasoning steps. Concretely, given math problems and labeled solution steps, we score each step to assess how well the scores correlate with the labels at the step and outcome granularity.

2. Inference-time interventions: we aim to evaluate how ProcessLID scores can be used as inference-time signals to guide LLM reasoning. Given a question, we generate multiple rollouts and use per-rollout scores as correctness signals for answer selection.

## 5.1 OFFLINE CORRECTNESS CLASSIFICATION ON LABELED COT

**Datasets** For step-level evaluation, we use **ProcessBench** (Zheng et al., 2025), which provides $\sim$3.4K reasoning trajectories with step-level labels from four math sources: GSM8K (Cobbe et al., 2021), MATH

|  | GSM8K | MATH | OmniMath | OlympiadBench |
|---|---|---|---|---|
| # of Trajectories | 375 | 955 | 951 | 561 |
| # of Steps | 1401 | 3969 | 4338 | 2731 |
| % of Correct Steps | 85.51 | 85.59 | 83.52 | 84.55 |

Table 1: Statistics of the sub-datasets of ProcessBench

(Hendrycks et al., 2021), OmniMath (Gao et al., 2024), and OlympiadBench (He et al., 2024), listed in order of increasing difficulty from grade school to Olympiad level. We partition ProcessBench into disjoint sub-datasets by question source and report results separately to make the effect of task difficulty explicit. To remove duplicated questions, we discard $\sim$600 trajectories, leaving $\sim$2.8K trajectories. Table 1 summarizes the resulting statistics.

For outcome-level evaluation, we sampled 5K problems from MATH and generated CoTs with step and outcome-level labels to curate **MATH-CoT-Labeled**; the final-answer accuracy is 77.98%.

**Models** We evaluate methods on six generator (base) models: Llama-3.1-8B-Instruct (Grattafiori et al., 2024) (abbrev. LLAMA-3.1-8B) and five DeepSeek-R1-Distill variants (DeepSeek-AI et al., 2025): DS-QW-1.5B, DS-QW-7B, DS-QW-14B, DS-QW-32B, and DS-LLAMA-8B. The DS-QW prefix models follows Qwen2.5 backbones (Qwen et al., 2025); DS-LLAMA-8B follows the Llama-3.1-8B backbone. For ProcessLID, We use DS-QW-14B as the prover for all base models.

**Evaluation Metrics** We report the Area Under the Receiver Operating Characteristic Curve (AU-ROC), which measures classification accuracy over various thresholds.

1. Step level: Let $j$ index a reasoning step and let $\mathcal{I}_j$ be the set of trajectories that contain a $j$-th step. For each $j$, we compute AUROC using scores $\{g(Q^{(i)}, S^{(i)}_{1:j}) : i \in \mathcal{I}_j\}$ and ground truths $\{\hat{y}_{i,j} : i \in \mathcal{I}_j\}$. We then average these per-step AUROC values over the set of valid step indices $\mathcal{J}$:

$$\text{AUC}_{\text{step}} = \frac{1}{|\mathcal{J}|} \sum_{j \in \mathcal{J}} \text{AUROC}\Big(\{g(Q^{(i)}, S^{(i)}_{1:j})\}_{i \in \mathcal{I}_j}, \{\hat{y}_{i,j}\}_{i \in \mathcal{I}_j}\Big).$$

   To ensure stability, $\mathcal{J}$ excludes step index with $< 50$ instances or labels from only one class.

2. Outcome level: We compare $g(Q^{(i)}, S^{(i)})$ for the final step of every trajectory with its ground truth $\hat{y}_{i,H_i}$ to compute a single AUROC:

$$\text{AUC}_{\text{outcome}} = \text{AUROC}\big(\{g(Q^{(i)}, S^{(i)})\}_{i=1}^{n}, \{\hat{y}_{i,H_i}\}_{i=1}^{n}\big).$$

**ProcessLID Setup** When computing the LID of either the base model $\pi$ or the prover model $\mu$ for a reasoning step $s_j$, we form the neighbor set from the $j$-th step of all other trajectories in the same dataset that contain a $j$-th step. We provide analysis on the choice of the neighborhood size in Appendix A.4. We choose the scalar weight $\alpha$ using a disjoint calibration split: we sweep $\alpha \in \{0.0, 0.2, 0.4, \ldots, 2.0\}$ and select the value that maximizes validation $\text{AUC}_{\text{step}}$.

**Baselines** We consider five baselines. *Random* assigns a score sampled from $\text{Unif}(0, 1)$. *LN Entropy* (Malinin & Gales, 2021) computes the length normalized entropy of a reasoning step by averaging token level entropies over the step (implementation in Appendix A.6). *CoT Entropy* (Ye et al., 2025b) approximates the entropy of the binary distribution on the correctness of a step. The approximation is done by marginalizing over $N$ sampled CoT from a judge LLM on whether the step is correct. We use DS-QW-14B as the judge LLM and sample $N = 10$ CoTs at high temperature per step. *Chain of Embeddings (CoE)* (Wang et al., 2025) averages over the embeddings of all tokens in a step at each layer of the LLM. An uncertainty score is computed based on the L2 norm difference and the angle between the aggregated embeddings at consecutive layers. *Vanilla LID* (Yin et al., 2024) directly computes the LID of a reasoning step using the base model, i.e., $m_\pi(E_{\pi,j})$.

Because *LN Entropy*, *CoE*, *Vanilla LID*, and ProcessLID only require representations from an LLM, when we "specify the generator model" we keep the input reasoning trajectories from the dataset fixed and vary only the LLM used to extract representations. This avoids the heavy computational cost of re-generating and re-labeling rollouts. In contrast, *Random* and *CoT Entropy* do not depend on the base model; therefore, they yield a single result across all base models per dataset.

Tables 2 summarize the benchmarks. Overall, ProcessLID is consistently the top performer across models and datasets, exceeding the strongest baselines by 7.33 AUROC at the step level and 9.83 AUROC at the outcome level. In addition, it delivers material gains over Vanilla LID, with a margin of 10.11 AUROC at the step level and 10.27 AUROC at the outcome level. These gains indicate that our innovations strengthen LID's discriminative power beyond its vanilla formulation.

**Robustness to task difficulty** As difficulty increases from grade school (GSM8K) to Olympiad (OlympiadBench) level, ProcessLID's step level AUROC varies by less than 2.41, indicating robustness across task difficulties. In contrast, CoT Entropy degrades monotonically from 88.83 to 77.78. This pattern reflects a fundamental limitation of external judgment methods: their performance is bounded by the external model's capability. Once task difficulty exceeds the external model's capability, the resulting reward is no longer reliable. In contrast, ProcessLID does not depend on inference time judgments from an external model, it provides a more stable internal reward when task difficulty varies widely.

**Robustness across base models** Within each dataset, ProcessLID varies by less than 5.31 AUROC across base models, suggesting minimal reliance on model-specific representation geometries. By

| Method / Model | LLAMA-3.1-8B | DS-QW-1.5B | DS-QW-7B | DS-LLAMA-8B | DS-QW-14B | DS-QW-32B | Average |
|---|---|---|---|---|---|---|---|
| ProcessBench GSM8K (Step-level) | | | | | | | |
| Random | - | - | - | - | - | - | 49.90 |
| CoT Entropy | - | - | - | - | - | - | 88.83 |
| LN Entropy | 70.49 | 72.64 | 75.30 | 70.72 | 75.70 | 71.87 | 72.79 |
| Chain of Embeddings | 58.76 | 69.98 | 73.49 | 67.85 | 64.54 | 74.14 | 68.12 |
| Vanilla LID | 82.15 | 77.56 | 81.30 | 74.12 | 83.55 | 78.59 | 79.55 |
| ProcessLID (Ours) | **90.88** | **89.95** | **89.22** | **90.08** | **92.22** | **91.09** | **90.57** |
| ProcessBench MATH (Step-level) | | | | | | | |
| Random | - | - | - | - | - | - | 49.50 |
| CoT Entropy | - | - | - | - | - | - | 83.31 |
| LN Entropy | 71.17 | 75.33 | 74.69 | 73.19 | 75.90 | 74.67 | 74.16 |
| Chain of Embeddings | 58.86 | 73.82 | 76.86 | 76.45 | 73.31 | 80.65 | 73.33 |
| Vanilla LID | 83.16 | 77.72 | 82.87 | 76.66 | 82.07 | 80.48 | 80.49 |
| ProcessLID (Ours) | **88.60** | **88.80** | **90.20** | **87.02** | **92.12** | **91.68** | **89.73** |
| ProcessBench OmniMath (Step-level) | | | | | | | |
| Random | - | - | - | - | - | - | 51.29 |
| CoT Entropy | - | - | - | - | - | - | 81.76 |
| LN Entropy | 65.90 | 70.49 | 69.67 | 69.07 | 70.71 | 70.52 | 69.44 |
| Chain of Embeddings | 59.63 | 70.07 | 71.79 | 75.82 | 71.74 | 75.87 | 70.82 |
| Vanilla LID | 83.14 | 78.69 | 82.93 | 77.91 | 80.52 | 80.84 | 80.67 |
| ProcessLID (Ours) | **88.35** | **88.04** | **88.72** | **85.79** | **90.21** | **90.33** | **88.57** |
| ProcessBench OlympiadBench (Step-level) | | | | | | | |
| Random | - | - | - | - | - | - | 50.36 |
| CoT Entropy | - | - | - | - | - | - | 77.78 |
| LN Entropy | 66.64 | 71.51 | 71.19 | 69.54 | 72.90 | 72.57 | 70.73 |
| Chain of Embeddings | 66.03 | 71.42 | 75.10 | 76.45 | 73.31 | 78.65 | 73.50 |
| Vanilla LID | 79.97 | 75.44 | 77.67 | 72.20 | 75.14 | 74.86 | 75.88 |
| ProcessLID (Ours) | **86.70** | **87.77** | **89.36** | **85.57** | **90.78** | **88.78** | **88.16** |
| MATH-CoT-Labeled (Outcome-level) | | | | | | | |
| Random | - | - | - | - | - | - | 49.24 |
| CoT Entropy | - | - | - | - | - | - | 69.01 |
| LN Entropy | 64.42 | 62.49 | 62.16 | 63.34 | 62.71 | 62.54 | 63.03 |
| Chain of Embeddings | 60.15 | 57.50 | 60.40 | 60.84 | 56.64 | 58.78 | 59.05 |
| Vanilla LID | 69.41 | 67.66 | 70.45 | 68.55 | 68.70 | 66.67 | 68.57 |
| ProcessLID (Ours) | **78.06** | **77.30** | **80.85** | **78.39** | **78.50** | **79.91** | **78.84** |

Table 2: Main results of step-level and outcome-level correctness characterization over four sub-datasets of ProcessBench. Best results are in **bold**.

comparison, CoE exhibits sharp drops (5.78–14.45 AUROC) on LLAMA-3.1-8B across all datasets. CoE relies on magnitude and angle features, assuming that correctness correlates with small angles and large magnitudes; such assumptions may not transfer across model architectures or training regimes. In contrast, ProcessLID computes LID from statistics that compare each step's representation relative to its neighborhood representations within the same base model, making no assumption about an absolute representation geometry.

Appendix A.7 reports the step-level AUROC computed by combining correctness signals from all steps and evaluating one AUROC (i.e., without first computing AUROC per step and then averaging). This result yields the same ranking and comparable margins.

## 5.2 INFERENCE-TIME INTERVENTION FOR ANSWER SELECTION

**Experiment setup** Let $\mathcal{D} = \{Q^{(1)}, \ldots, Q^{(M)}\}$ be the questions with ground truth answers $\mathcal{A} = \{A^{(1)}, \ldots, A^{(M)}\}$. For each $Q^{(i)}$, we sample $N$ CoT trajectories $S^{(i,1)}, \ldots, S^{(i,N)}$ from the generator model with answers $\hat{A}^{(i,1)}, \ldots, \hat{A}^{(i,N)}$. For each rollout $S^{(i,n)}$, we assign $g(Q^{(i)}, S^{(i,n)})$ as the rollout's correctness signal, i.e., scores assigned by ProcessLID or other baselines. We set $N = 10$ and report results averaged over three independent trials with different random seeds.

**Datasets and Models** We use LLAMA-3.1-8B and DS-LLAMA-8B as our generator model $\pi$. We randomly sample 150 questions from the dataset **MATH-500** (Lightman et al., 2023).

**ProcessLID setup** We segment each rollout into semantically self-contained steps by splitting on double-newline delimiters (\n\n) and take the final step's score $R(E_{\pi,H})$ as the rollout signal. Because $R(E_{\pi,j})$ depend on $m_\pi(E_{\pi,j-1})$ and $E_{\mu,j-1}$ from the previous step $s_{j-1}$, we compute scores sequentially for all steps $j = 1, \ldots, H$.

**Baselines** We consider four baselines. *Random* assigns each rollout a score sampled from Unif$(0, 1)$. *LN Entropy* computes the same length-normalized entropy described in Section 5.1 for each rollout. *Vanilla LID* computes the same LID described in Section 5.1 and follows the same setup as ProcessLID described in the previous section. *Reward Model* is a sequence scorer that assigns higher values to responses better aligned with a given prompt; we use Skywork-Reward-V2-Llama-3.1-8B (Liu et al., 2025), a LLAMA-3.1-8B model with a scalar head fine-tuned on preference data. It is ranked first on RewardBench V2 (Malik et al., 2025), a comprehensive benchmark evaluating reward models' alignment with human preferences. Because this model is a high-capacity generative backbone explicitly trained to score sequences, it serves as a strong upper-bound across evaluation metrics.

**Evaluation metric** We consider three accuracy metrics.

1. Oracle@N (Pruned-5): remove the five candidates with the smallest $g(Q^{(i)}, S^{(i,n)})$. We consider the model correct if any of the remaining $N - 5$ candidates is correct.

2. Best-of-N (BoN): select $\hat{A}^{(i,*)} = \arg\max_n g(Q^{(i)}, S^{(i,n)})$. We consider the model correct iff $\hat{A}^{(i,*)}$ is correct.

3. Weighted Majority Vote: assign to each candidate $\hat{A}^{(i,n)}$ a weight equal to the Softmax-normalized value of $g(Q^{(i)}, S^{(i,n)})$ with temperature $\tau = 5$. Perform a weighted majority vote over candidates. We consider the model correct if the selected candidate is correct.

| Method / Model | LLAMA-3.1-8B | DS-LLAMA-8B | *Average* |
|---|---|---|---|
| Oracle @ N Pruned 5 | | | |
| Reward Model | $75.56_{2.33}$ | $82.44_{1.28}$ | 79.00 |
| Random | $71.78_{1.58}$ | $79.44_{1.31}$ | 75.61 |
| LN Entropy | $71.11_{1.55}$ | $79.11_{2.16}$ | 75.11 |
| Vanilla LID | $75.33_{1.07}$ | $79.33_{4.65}$ | 77.33 |
| ProcessLID (Ours) | $\mathbf{78.00_{3.20}}$ | $\mathbf{84.00_{1.85}}$ | **81.00** |
| Weighted Majority Vote | | | |
| Reward Model | $63.56_{2.85}$ | $\mathbf{80.44_{0.71}}$ | 72.00 |
| Random | $62.00_{1.26}$ | $79.22_{1.17}$ | 70.61 |
| LN Entropy | $62.89_{0.36}$ | $78.89_{0.94}$ | 70.89 |
| Vanilla LID | $64.67_{2.82}$ | $78.67_{2.82}$ | 71.67 |
| ProcessLID (Ours) | $\mathbf{65.33_{3.20}}$ | $79.33_{1.85}$ | **72.33** |
| Best of N | | | |
| Reward Model | $\mathbf{59.11_{1.42}}$ | $\mathbf{76.44_{1.78}}$ | **67.78** |
| Random | $49.16_{0.62}$ | $63.18_{0.29}$ | 56.17 |
| LN Entropy | $46.10_{3.26}$ | $64.44_{2.33}$ | 55.22 |
| Vanilla LID | $56.67_{3.24}$ | $65.33_{3.85}$ | 61.00 |
| ProcessLID (Ours) | $57.33_{2.13}$ | $74.59_{2.97}$ | 65.96 |

Table 3: Main results of rollout. The presented accuracies are averaged over three independent trials. Subscript values represent the 95% confidence interval of the mean accuracy. Best results are in **bold**. Second-to-best results are in underline

Table 3 presents the results. On *Oracle@N (Pruned-5)* and *Weighted Majority Vote*, our method outperforms the reward model for both generator models by an average of 2.00 and 0.33 AUROC. For *Best of N*, our method trails the reward model by only 1.82 AUROC on average. Surpassing the reward model on two metrics and remaining highly competitive on *Best of N*—all without any additional training or task-specific supervision—underscores the practicality of ProcessLID.

We also highlight that our method outperforms Vanilla LID on all three metrics by average margins of 3.67, 0.66, and 4.96 AUROC. This demonstrates a substantial improvement of ProcessLID over its vanilla counterpart.

# 6 ANALYSIS

## 6.1 COMPONENT ABLATION

Our **ProcessLID** has three essential components: (i) a prover model $\mu$ that evaluates LID under a distinct representation geometry, (ii) two asymmetric delta signals (Step LID Delta and LID Embedding Delta) that isolate the correctness of a single step, and (iii) aggregation across layers of the LLM that captures strong signals from multiple layers. We ablate each component under the same experimental setup as in Section 5.1 by evaluating the following variants:

1. *w/o Aggregation* replaces Fisher's aggregation over $\mathcal{L}^*$ with a single best layer;

2. *Vanilla w/ Prover* computes vanilla LID for both base and prover terms: $R(E_{\pi,j}) = m_\pi(E_{\pi,j}) + \alpha m_\mu(E_{\mu,j})$;

3. *Only Prover* uses only the prover term: $R(E_{\pi,j}) = m_\mu(\Delta E_{\mu,j})$;

4. *Reversed* swaps the roles of base and prover: $R(E_{\pi,j}) = \Delta m_\mu(E_{\mu,j}) + \alpha m_\pi(\Delta E_{\pi,j})$.

| Method / Model | LLAMA-3.1-8B | DS-QW-1.5B | DS-QW-7B | DS-LLAMA-8B | DS-QW-14B | DS-QW-32B | *Average* |
|---|---|---|---|---|---|---|---|
| ProcessBench GSM8K | | | | | | | |
| ProcessLID (Original) | 90.88 | 89.95 | 89.22 | 90.08 | **92.22** | 91.09 | 90.57 |
| w/o Aggregation | 89.72 | 89.41 | 88.41 | 88.58 | 90.51 | 89.55 | 89.36 ($-1.21$) |
| Reversed | 88.43 | 88.24 | 86.99 | 87.34 | 89.61 | 88.72 | 88.22 ($-2.35$) |
| Only Prover | 87.25 | 86.10 | 86.10 | 87.25 | 88.41 | 88.41 | 87.25 ($-3.20$) |
| Vanilla w/ Prover | **91.43** | **91.33** | **90.96** | **91.87** | 91.91 | **91.60** | **91.52 ($+1.03$)** |
| ProcessBench MATH | | | | | | | |
| ProcessLID (Original) | **88.60** | **88.80** | **90.20** | **87.02** | 92.12 | 91.68 | **89.73** |
| w/o Aggregation | 86.92 | 88.04 | 88.52 | 86.68 | 90.13 | 90.23 | 88.42 ($-1.31$) |
| Reversed | 85.12 | 86.44 | 87.29 | 85.09 | 89.58 | 89.08 | 87.10 ($-2.63$) |
| Only Prover | 86.83 | 86.96 | 86.96 | 86.83 | 87.42 | 87.42 | 87.07 ($-2.66$) |
| Vanilla w/ Prover | 87.71 | 86.90 | 87.95 | 85.28 | 86.08 | 88.01 | 86.99 ($-2.74$) |
| ProcessBench OmniMath | | | | | | | |
| ProcessLID (Original) | **88.35** | **88.04** | **88.72** | **85.79** | 90.21 | 90.33 | **88.57** |
| w/o Aggregation | 87.61 | 86.81 | 86.73 | 85.09 | 89.28 | 88.80 | 87.39 ($-1.18$) |
| Reversed | 85.12 | 86.44 | 87.29 | 85.09 | 89.68 | 89.08 | 87.12 ($-1.45$) |
| Only Prover | 83.34 | 84.35 | 84.35 | 83.84 | 86.66 | 86.66 | 84.87 ($-3.70$) |
| Vanilla w/ Prover | 85.03 | 83.43 | 84.84 | 81.35 | 79.10 | 84.70 | 83.08 ($-5.49$) |
| ProcessBench OlympiadBench | | | | | | | |
| ProcessLID (Original) | 86.70 | **87.77** | **89.36** | 85.57 | 90.78 | 88.78 | **88.16** |
| w/o Aggregation | **87.27** | 87.52 | 88.28 | 84.23 | 90.62 | 88.46 | 87.73 ($-0.43$) |
| Reversed | 85.80 | 85.26 | 86.55 | 81.77 | 90.17 | 86.67 | 86.04 ($-2.12$) |
| Only Prover | 86.83 | 86.49 | 86.49 | 86.83 | 87.92 | 87.92 | 87.08 ($-1.08$) |
| Vanilla w/ Prover | 85.73 | 81.80 | 84.37 | 81.54 | 79.59 | 83.87 | 82.82 ($-5.34$) |

Table 4: Component ablation results over four sub-datasets of ProcessBench. Value in parentheses under the *Average* column indicates the performance difference with ProcessLID. Best results are in **bold**

Table 4 summarizes the results. In 19 out of 24 settings, ProcessLID outperforms all ablated variants; in the remaining cases, the gap is small ($< 1.8$ AUROC), suggesting there is no systematic setting in which a simpler variant is clearly preferable.

The *Reversed* variant exhibits an average drop of 2.13 AUROC, underscoring that ProcessLID's specific asymmetric formulation is not arbitrary: letting the base model contribute the Step LID Delta and the prover contribute the LID Embedding Delta yields noticeably stronger signals than swapping their roles (see also Section 6.4 for a deeper analysis).

Moreover, the *Vanilla w/ Prover* variant (average drop 3.14 AUROC) and the *Only Prover* variant (average drop 2.66 AUROC) show that our gains are not simply due to adding a stronger prover model. *Vanilla w/ Prover* has access to exactly the same base and prover representations as ProcessLID, but lacks the delta-based design and layer-wise aggregation; its weaker performance indicates that the prover model's raw representation power alone is insufficient. Likewise, *Only Prover* demonstrates that the prover signal by itself cannot match ProcessLID, highlighting the importance of the asymmetric synergy between the base model's Step LID Delta $\Delta m_\pi(E_{\pi,j})$ and the prover's LID Embedding Delta $m_\mu(\Delta E_{\mu,j})$.

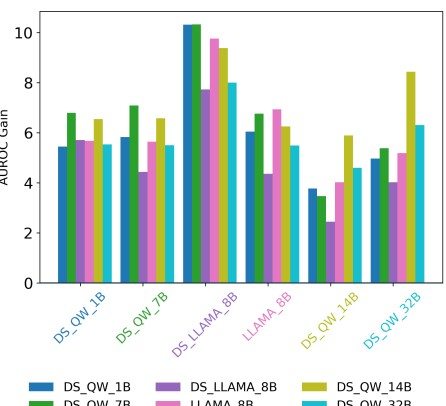

Figure 2: AUROC improvement from adding a prover $\mu$, relative to the *w/o Prover* variant. Bars are grouped by base model $\pi$ (x-axis). Within each base model, bars correspond to candidate provers (legend), ordered by increasing parameter size.

Overall, these ablations indicate that all three ingredients of ProcessLID—the prover's distinct representation geometry, the asymmetric formulation, and aggregating signals across layers—each contribute substantially to the final performance, and that the improvement over vanilla LID is not dominated by merely using a stronger prover.

## 6.2 PROVER MODEL ABLATION

We analyze how the choice of prover $\mu$ affects ProcessLID. For each base model $\pi$ described in Section 5.1, we set its prover $\mu$ to every other base models, compute the step-level AUROC of

$R(E_{\pi,j})$, and report the gain over the *w/o Prover* variant (Section 6.1). This isolates the contribution of the prover itself.

Figure 2 shows results averaged across the four ProcessBench subsets; per-subset results are provided in Appendix A.8. We observe a consistent non-monotonic pattern: gains do not scale with prover size. For all base models, the performance gain first increases as $\mu$ scales, achieves the best performance gain for a prover that is moderately weaker or stronger than $\pi$ (e.g., DS–QW–14B in many cases), and then declines for the larger provers. This contrasts with external-judgment approaches (judge LLMs or PRMs), where the reward reliability typically scales with the external model's size.

## 6.3 ANALYSIS ON THE EFFECTIVENESS OF PROVER MODELS

Motivated by the non-monotonic relationship between prover size and performance gains observed in Section 6.2, we ask how and why a prover helps. We hypothesize the prover's effectiveness is governed by two axes: (i) Alignment—how much correctness information in the prover's signal is not captured by the base signal; and (ii) Separability—the prover signal's own discriminative power.

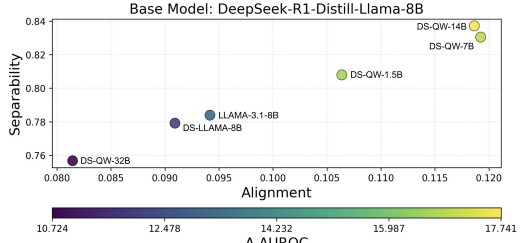

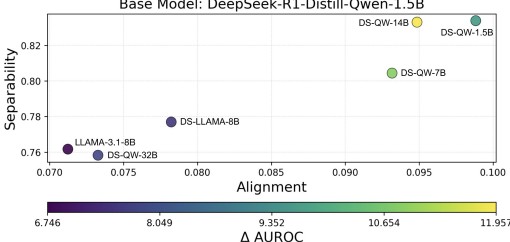

Figure 3: AUROC gain for candidate prover models $\mu$ as a function of Alignment (x-axis) and Separability (y-axis), with the base model fixed. Each point is labeled by $\mu$; point color indicates the AUROC gain contributed by that prover. The color bar maps from color to the numeric AUROC gain (darker = smaller, brighter = larger).

**Alignment** We define Alignment as the mutual information $I(\mathbf{r}; \hat{\mathbf{Y}})$ between the residual $\mathbf{r}$ and the ground-truth step labels $\hat{\mathbf{Y}}$. Here, $\mathbf{r}$ is the prover signal's "new" information not captured by the base model. We obtain $\mathbf{r}$ by removing from the prover's signal whatever can be explained by the base model's signal via kernel regression. Larger values of $I(\mathbf{r}; \hat{\mathbf{Y}})$ indicate that the prover contributes more label-relevant (correctness-relevant) information beyond the base model. See Appendix A.9 for a formal construction and validation.

**Separability** We measure Separability as the step-level AUROC (Section 5.1) when scoring steps only with the prover's signal $m_\mu(\Delta E_{\mu,j})$.

For each base model $\pi$ (described in Section 5.1), we set its prover $\mu$ to every other base model and evaluate on ProcessBench using the step-level experiment setup in Section 5.1. For each $(\pi, \mu)$ pair, we report the AUROC improvement obtained by using ProcessLID as the correctness signal relative to using the base signal alone ($m_\pi(E_{\pi,j})$); this isolates the performance gain attributable to the prover.

Figure 3 shows a clear trend: as both Separability and Alignment increase, the AUROC gain from the prover also increases. The same pattern holds for other base models (Appendix A.11). This explains the non-monotonic trend in Section 6.2: the largest prover (e.g., DeepSeek–Qwen–32B) often exhibits lower Separability and Alignment, suggesting larger size does not guarantee complementary, label-relevant correctness information. In short, effective prover selection should prioritize models that jointly maximize Separability and Alignment rather than raw size.

## 6.4 ANALYSIS ON THE ASYMMETRIC FORMULATION OF PROCESSLID

Section 6.3 shows that a prover helps the most when it has high alignment and separability. We argue the asymmetric design $R_j = \Delta m_\pi(E_{\pi,j}) + \alpha m_\mu(\Delta E_{\mu,j})$ is meant to exploit this: the base model contributes a Step LID Delta, while the prover contributes the LID of its embedding displacement. This pairing reduces redundancy between signals and preserves the prover's complementary discriminative power.

| Formulation / Model | LLAMA-3.1-8B | DS-QW-1.5B | DS-QW-7B | DS-LLAMA-8B | DS-QW-14B | DS-QW-32B | *Average* |
|---|---|---|---|---|---|---|---|
| | ProcessBench (Alignment ↑ / AUROC ↑) | | | | | | |
| Both Step LID Delta | 0.077 / 86.94 | 0.067 / 82.14 | 0.069 / 82.77 | 0.079 / 79.64 | 0.091 / 80.11 | 0.074 / 83.95 | 0.076 / 82.59 |
| Both LID Embedding Delta | **0.099** / 85.97 | 0.073 / 86.03 | 0.068 / 86.70 | 0.095 / 84.37 | **0.11** / 85.81 | 0.10 / 86.03 | 0.092 / 85.82 |
| ProcessLID (Original) | 0.078 / **89.41** | **0.095 / 87.03** | **0.095 / 87.73** | **0.12** / 85.36 | 0.10 / **89.74** | **0.10 / 89.46** | **0.10 / 88.12** |

Table 5: Alignment and AUROC results for ProcessLID and its symmetric variants, evaluated over ProcessBench. Best results are in **bold**.

We compare ProcessLID's formulation to two symmetric alternatives: *Both Step LID Delta* $\Delta m_\pi(E_{\pi,j}) + \alpha \, \Delta m_\mu(E_{\mu,j})$, and *Both LID Embedding Delta* $m_\pi(\Delta E_{\pi,j}) + \alpha \, m_\mu(\Delta E_{\mu,j})$ using the step-level experiment setup described in Section 5.1. Table 5 reports, for each base model evaluated on ProcessBench, the step-level performance and Alignment $I(\mathbf{r}; \hat{\mathbf{Y}})$.

Across nearly all base models, the asymmetric ProcessLID achieves the highest Alignment and the best AUROC, followed by *Both LID Embedding Delta*, then *Both Step LID Delta*. These results suggest that combining a base-model Step LID Delta with a prover LID Embedding Delta maximizes the prover's unique, correctness-relevant contribution, yielding the strongest performance.

# 7 CONCLUSION

In this paper, we proposed a training-free, step-level internal reward signal. Our method achieves state-of-the-art performance across four datasets and six models, and remains competitive under inference-time scenarios. Ablation and alignment–separability analyses validated each component of ProcessLID and revealed why the prover model and asymmetric formulation are especially effective.

## ETHICS STATEMENT

We acknowledge and adhere to the ICLR Code of Ethics. This research uses publicly available mathematics datasets and pretrained models; it does not involve human subjects, the collection of personal data, or deployment in sensitive settings. No protected attributes are inferred or utilized. To the best of our knowledge, the work raises no concerns regarding privacy, safety, fairness, legal compliance, or research integrity, and it does not violate the ICLR Code of Ethics.

## REPRODUCIBILITY STATEMENT

To facilitate reproducibility of our results, we provide detailed documentation of the full experimental setup—models, datasets, baselines, and evaluation protocols—in Section 5. The detailed procedures for prompt formatting, embedding extraction, and inference-time rollouts are described in Appendix A.5. Relevant implementation details of ProcessLID are provided in Appendix A.2.

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

# A APPENDIX

## A.1 USAGE OF LLMs

We used an LLM (ChatGPT) solely to assist with copy-editing this manuscript (correcting typos, improving grammar, and rephrasing sentences for clarity and fluency). The LLM did not contribute to research ideation, problem formulation, experimental design, data collection, implementation, analysis, or interpretation of results; all technical content and decisions were made by the authors.

## A.2 AGGREGATING PROCESSLID ACROSS LAYERS

**Setup and notation** For each step index $j$ and layer $l \in \{1, \ldots, L\}$, let $R(E_{\pi,j,l})$ denote the ProcessLID score evaluated at the last token of step $s_j$ using the representation from layer $l$ (as defined in the main text). For calibration, we assume a disjoint set

$$\mathcal{C}_{l,j} = \{(R(E_{\pi,j,l}^{(m)}), \hat{y}_j^{(m)})\}_{m=1}^{M},$$

available for each $(l, j)$, where $\hat{y}_j^{(m)} \in \{0, 1\}$ is the step-level ground-truth label ($1 = $ correct).

**P-value normalization** We cast layer-wise scoring as a one-sided test "is this step incorrect?" against the null "this step looks like a correct one." Formally, let $F_{l,j}^{+}$ be the empirical distribution of $R(\cdot)$ over *correct* steps in the calibration set at layer $l$, step $j$, we define the null hypothesis

$$H_{0,l,j} : \ R(E_{\pi,j,l}) \text{ is typical of correct steps (drawn from } F_{l,j}^{+})$$

and the alternative hypothesis

$$H_{1,l,j} : \ R(E_{\pi,j,l}) \text{ is larger than typical (evidence of incorrectness)}$$

With the convention that larger $R$ indicates stronger evidence of incorrectness, we compute an upper-tail $p$-value:

$$p_{\pi,j,l}^{(i)} = \mathbb{P}_{T \sim F_{l,j}^{+}}[T \geq R(E_{\pi,j,l}^{(i)})] \approx \frac{1}{|\mathcal{C}_{l,j}^{+}|} \sum_{(R', \hat{y}') \in \mathcal{C}_{l,j}^{+}} \mathbf{1}\Big[R' \geq R(E_{\pi,j,l}^{(i)})\Big]$$

where $\mathcal{C}_{l,j}^{+} = \{(R', \hat{y}') \in \mathcal{C}_{l,j} : \hat{y}' = 1\}$.

**Selecting a subset of layers $\mathcal{L}^*$ (greedy selection on calibration)** Aggregating *all* layers empirically under-performs a single best layer because layers differ in how informative their correctness signals are. We therefore select a subset $\mathcal{L}^* \subseteq \{1, \ldots, L\}$ on the calibration split:

1. *Rank layers by individual quality.* For each layer $l$, compute AUROC on $\mathcal{C}_{l,j}$ using $R(E_{\pi,j,l})$ as the correctness signal, then rank layers $l_1, \ldots, l_L$ by decreasing AUROC.

2. *Evaluate top-$k$ aggregates.* For $k = 1, \ldots, L$:

   (a) First convert each per-layer score to a $p$-value using the procedure above, obtaining $\{p_{\pi,j,l}^{(m)}\}$ for all layers.

   (b) For every calibration example $(m, j)$, aggregate the $p$-values from the top-$k$ layers via Fisher's method:
   $$T_j^{(m)}(k) = -2 \sum_{r=1}^{k} \ln(p_{\pi,j,l_r}^{(m)}).$$

   (c) Compute AUROC of $T_j^{(m)}(k)$ against the labels $\hat{y}_j^{(m)}$.

3. *Choose the subset.* Let $k^* = \arg\max_k \text{AUROC}(T(\cdot; k))$ (breaking ties by the smallest $k$), and set $\mathcal{L}^* = \{l_1, \ldots, l_{k^*}\}$.

At test time, we convert per-layer scores to $p$-values as above and report the Fisher aggregate

$$T_j = -2 \sum_{l \in \mathcal{L}^*} \ln(p_{\pi,j,l}).$$

Larger $T_j$ indicates stronger evidence that step $s_j$ is incorrect. Implementation details for $p$-value construction and the selection routine are provided here for reproducibility; the same calibration split is used both to learn $\mathcal{L}^*$ and to determine the null distributions $F_{l,j}^{+}$.

### A.3 ROBUSTNESS ANALYSIS ON HYPERPARAMETER CALIBRATION

ProcessLID relies on a calibration dataset to set two hyperparameters:

1. The positive scalar $\alpha$ that weights the base and prover terms in ProcessLID,

$$R\big(E_{\pi,j}\big) \;=\; \Delta m_\pi\big(E_{\pi,j}\big) \;+\; \alpha\, m_\mu\big(\Delta E_{\mu,j}\big),$$

2. The subset of best-performing layers $\mathcal{L}^*$ used in Fisher's method to aggregate per-layer ProcessLID scores into a single value.

Using the same calibration set, ProcessLID also normalizes the score at each layer into a layer-wise $p$-value before applying Fisher's method.

To assess the robustness of these hyperparameters and the $p$-value normalization procedure, we study (i) how performance varies as we vary the size of the calibration dataset, and (ii) how well the calibrated hyperparameters transfer across different datasets and base models.

#### A.3.1 IMPACT OF CALIBRATION DATASET SIZE ON PROCESSLID PERFORMANCE

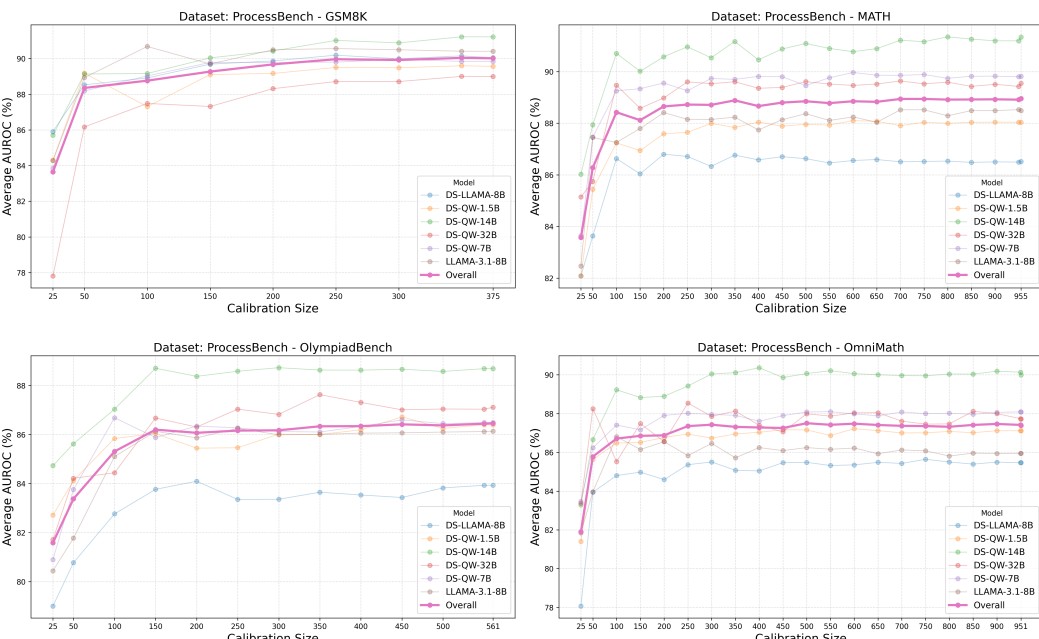

Figure 4: Calibration dataset size vs. ProcessLID AUROC. For each ProcessBench sub-dataset, we vary the calibration dataset size and plot the average ProcessLID step-level performance with the hyperparameter calibrated at the specified dataset size Semi-transparent lines show per-base-model curves; solid lines are averages over all six base models.

In this section, we evaluate how the size of the calibration dataset affects the performance of ProcessLID on the four ProcessBench sub-datasets. For each dataset, we vary the calibration size over $\{25, 50\}$ and then in increments of 50 up to the full dataset, and measure the step-level performance of each base model under the experimental setup described in Section 5.1.

Figure 4 summarizes the results. Across all four sub-datasets, performance quickly stabilizes once the calibration set contains roughly 150–200 data points, and noticeable degradations only occur when the calibration size drops below 100–150 data points. This indicates that our hyperparameter calibration and $p$-value normalization procedure is highly sample-efficient, requiring as few as 150–200 examples to reach near-optimal performance.

#### A.3.2 TRANSFERABILITY OF CALIBRATED HYPERPARAMETERS

We first evaluate how well the calibrated hyperparameters transfer across the sub-datasets in ProcessBench. For each base model and source dataset, we calibrate the hyperparameters on that dataset

| | ProcessBench - GSM8K | ProcessBench - MATH | ProcessBench - OlympiadBench | ProcessBench - OmniMath |
|---|---|---|---|---|
| ProcessBench - GSM8K | 90.05 | 86.50 | 87.86 | 86.79 |
| ProcessBench - MATH | 77.41 | 86.51 | 82.13 | 81.87 |
| ProcessBench - OlympiadBench | 76.74 | 76.78 | 83.93 | 81.31 |
| ProcessBench - OmniMath | 76.56 | 77.05 | 81.08 | 85.46 |

(a) DS-LLAMA-8B

| | ProcessBench - GSM8K | ProcessBench - MATH | ProcessBench - OlympiadBench | ProcessBench - OmniMath |
|---|---|---|---|---|
| ProcessBench - GSM8K | 89.56 | 86.84 | 83.04 | 86.71 |
| ProcessBench - MATH | 81.53 | 88.03 | 82.74 | 85.80 |
| ProcessBench - OlympiadBench | 80.08 | 80.21 | 86.40 | 82.79 |
| ProcessBench - OmniMath | 79.44 | 81.71 | 80.97 | 87.11 |

(b) DS-QW-1.5B

| | ProcessBench - GSM8K | ProcessBench - MATH | ProcessBench - OlympiadBench | ProcessBench - OmniMath |
|---|---|---|---|---|
| ProcessBench - GSM8K | 89.84 | 83.78 | 84.02 | 81.81 |
| ProcessBench - MATH | 78.03 | 89.82 | 84.83 | 83.75 |
| ProcessBench - OlympiadBench | 77.99 | 82.16 | 86.49 | 82.06 |
| ProcessBench - OmniMath | 80.38 | 79.45 | 83.05 | 88.09 |

(c) DS-QW-7B

| | ProcessBench - GSM8K | ProcessBench - MATH | ProcessBench - OlympiadBench | ProcessBench - OmniMath |
|---|---|---|---|---|
| ProcessBench - GSM8K | 91.22 | 88.48 | 85.23 | 87.96 |
| ProcessBench - MATH | 83.65 | 91.33 | 85.69 | 87.08 |
| ProcessBench - OlympiadBench | 80.49 | 81.94 | 88.69 | 82.53 |
| ProcessBench - OmniMath | 79.64 | 81.18 | 84.95 | 89.99 |

(d) DS-QW-14B

| | ProcessBench - GSM8K | ProcessBench - MATH | ProcessBench - OlympiadBench | ProcessBench - OmniMath |
|---|---|---|---|---|
| ProcessBench - GSM8K | 88.99 | 85.09 | 78.77 | 83.80 |
| ProcessBench - MATH | 81.98 | 89.55 | 82.77 | 80.32 |
| ProcessBench - OlympiadBench | 76.62 | 82.47 | 87.11 | 81.55 |
| ProcessBench - OmniMath | 78.59 | 80.77 | 82.79 | 87.72 |

(e) DS-QW-32B

| | ProcessBench - GSM8K | ProcessBench - MATH | ProcessBench - OlympiadBench | ProcessBench - OmniMath |
|---|---|---|---|---|
| ProcessBench - GSM8K | 90.41 | 89.02 | 83.94 | 87.27 |
| ProcessBench - MATH | 79.75 | 88.49 | 82.92 | 82.04 |
| ProcessBench - OlympiadBench | 77.41 | 82.22 | 86.13 | 82.76 |
| ProcessBench - OmniMath | 75.03 | 78.44 | 78.14 | 85.95 |

(f) LLAMA-3.1-8B

| | ProcessBench - GSM8K | ProcessBench - MATH | ProcessBench - OlympiadBench | ProcessBench - OmniMath |
|---|---|---|---|---|
| ProcessBench - GSM8K | 90.01 | 86.62 | 83.81 | 85.72 |
| ProcessBench - MATH | 80.39 | 88.96 | 83.51 | 83.48 |
| ProcessBench - OlympiadBench | 78.22 | 80.96 | 86.46 | 82.17 |
| ProcessBench - OmniMath | 78.27 | 79.77 | 81.83 | 87.39 |

(g) Overall

Table 6: Results of hyperparameter transferability across datasets. Each sub-table corresponds to one base-model. Row names indicate the dataset on which the hyperparameters are calibrated, and column names indicate the dataset to which those hyperparameters are transferred.

and then transfer them to all other datasets with the same base model, measuring the step-level performance under the experimental setup described in Section 5.1. The detailed results are reported in Table 6.

We then evaluate how well the calibrated hyperparameters transfer across base models. For each dataset and source model, we calibrate the hyperparameters on that model and then transfer them to all other base models with the same dataset, again using the step-level setup in Section 5.1. The corresponding results are shown in Table 7.

From Table 6g and Table 7e, which report results averaged over all base models and all datasets respectively, we observe that transferring hyperparameters across datasets leads to an average performance drop of only 6.14 AUROC, while transferring them across models leads to an average drop of 4.62 AUROC. In both cases, the degradation is small, indicating that our calibrated hyperparameters are highly robust to changes in both datasets and base models.

| | DS-LLAMA-8B | DS-QW-1.5B | DS-QW-14B | DS-QW-32B | DS-QW-7B | LLAMA-3.1-8B |
|---|---|---|---|---|---|---|
| DS-LLAMA-8B | 90.05 | 84.46 | 88.26 | 85.89 | 87.01 | 88.28 |
| DS-QW-1.5B | 83.94 | 89.56 | 83.92 | 81.18 | 84.66 | 87.35 |
| DS-QW-14B | 85.96 | 82.92 | 91.22 | 85.30 | 81.14 | 86.55 |
| DS-QW-32B | 89.26 | 84.06 | 89.87 | 88.99 | 86.79 | 88.38 |
| DS-QW-7B | 86.74 | 84.82 | 85.72 | 83.06 | 89.84 | 85.81 |
| LLAMA-3.1-8B | 85.93 | 84.38 | 84.37 | 82.97 | 81.71 | 90.41 |

(a) ProcessBench - GSM8K

| | DS-LLAMA-8B | DS-QW-1.5B | DS-QW-14B | DS-QW-32B | DS-QW-7B | LLAMA-3.1-8B |
|---|---|---|---|---|---|---|
| DS-LLAMA-8B | 86.51 | 84.74 | 86.41 | 86.96 | 87.07 | 84.42 |
| DS-QW-1.5B | 83.41 | 88.03 | 83.76 | 85.08 | 88.41 | 83.98 |
| DS-QW-14B | 83.90 | 79.46 | 91.33 | 86.90 | 82.99 | 84.61 |
| DS-QW-32B | 84.06 | 83.46 | 87.95 | 89.55 | 87.85 | 84.73 |
| DS-QW-7B | 81.14 | 85.54 | 82.74 | 83.35 | 89.82 | 83.27 |
| LLAMA-3.1-8B | 83.39 | 78.78 | 86.74 | 85.32 | 85.96 | 88.49 |

(b) ProcessBench - MATH

| | DS-LLAMA-8B | DS-QW-1.5B | DS-QW-14B | DS-QW-32B | DS-QW-7B | LLAMA-3.1-8B |
|---|---|---|---|---|---|---|
| DS-LLAMA-8B | 83.93 | 83.03 | 84.32 | 83.99 | 85.45 | 82.66 |
| DS-QW-1.5B | 79.22 | 86.40 | 80.03 | 81.71 | 83.40 | 82.50 |
| DS-QW-14B | 78.88 | 77.10 | 88.69 | 84.67 | 84.12 | 83.56 |
| DS-QW-32B | 78.99 | 77.49 | 85.91 | 87.11 | 82.52 | 84.97 |
| DS-QW-7B | 76.94 | 80.45 | 81.25 | 82.35 | 86.49 | 81.71 |
| LLAMA-3.1-8B | 75.96 | 80.32 | 81.00 | 79.68 | 82.34 | 86.13 |

(c) ProcessBench - OlympiadBench

| | DS-LLAMA-8B | DS-QW-1.5B | DS-QW-14B | DS-QW-32B | DS-QW-7B | LLAMA-3.1-8B |
|---|---|---|---|---|---|---|
| DS-LLAMA-8B | 85.46 | 81.98 | 85.46 | 84.71 | 86.50 | 83.72 |
| DS-QW-1.5B | 81.69 | 87.11 | 84.64 | 82.80 | 84.81 | 82.17 |
| DS-QW-14B | 78.73 | 79.62 | 89.99 | 86.05 | 83.79 | 82.97 |
| DS-QW-32B | 81.92 | 77.40 | 87.74 | 87.72 | 84.71 | 83.71 |
| DS-QW-7B | 79.93 | 82.05 | 84.71 | 83.31 | 88.09 | 82.30 |
| LLAMA-3.1-8B | 79.26 | 81.84 | 82.57 | 81.91 | 82.34 | 85.95 |

(d) ProcessBench - Omnimath

| | DS-LLAMA-8B | DS-QW-1.5B | DS-QW-14B | DS-QW-32B | DS-QW-7B | LLAMA-3.1-8B |
|---|---|---|---|---|---|---|
| DS-LLAMA-8B | 86.49 | 83.55 | 86.11 | 85.39 | 86.50 | 84.77 |
| DS-QW-1.5B | 82.07 | 87.78 | 83.09 | 82.69 | 85.32 | 84.00 |
| DS-QW-14B | 81.87 | 79.78 | 90.31 | 85.73 | 83.01 | 84.42 |
| DS-QW-32B | 83.56 | 80.60 | 87.87 | 88.34 | 85.47 | 85.45 |
| DS-QW-7B | 81.19 | 83.22 | 83.61 | 83.02 | 88.56 | 83.27 |
| LLAMA-3.1-8B | 81.13 | 81.33 | 83.67 | 82.47 | 83.09 | 87.74 |

(e) Overall

Table 7: Results of hyperparameter transferability across models. Each sub-table corresponds to one ProcessBench sub-dataset. Row names indicate the model on which the hyperparameters are calibrated, and column names indicate the model to which those hyperparameters are transferred.

A.4 ANALYSIS ON THE NEIGHBORHOOD SIZE IN LID COMPUTATION

**Setup** To study how the neighborhood size $T$ affects both the LID statistic and downstream performance, we sweep $T$ from 10 up to the number of trajectories available at a given step index within each ProcessBench sub-dataset (see row "# of Trajectories" in Table 1 for maximum neighborhood size). For each $T$, and for each base model $\pi$ (with the prover $\mu$ fixed to DS–QW–14B) described in Section 5.1, we compute: (i) the mean base-model LID $m_\pi(E)$ and (ii) the ProcessLID step-level AUROC.

**Findings** Figure 5 presents the results. Across all four subsets, the base-model LID $m_\pi(E)$ shows a characteristic pattern as $T$ grows: a sharp initial decrease followed by a plateau once $T$ reaches a sufficiently large neighborhood size. The ProcessLID AUROC exhibits the complementary trend: an initial increase followed by a plateau. This alignment suggests that once the local neighbor-

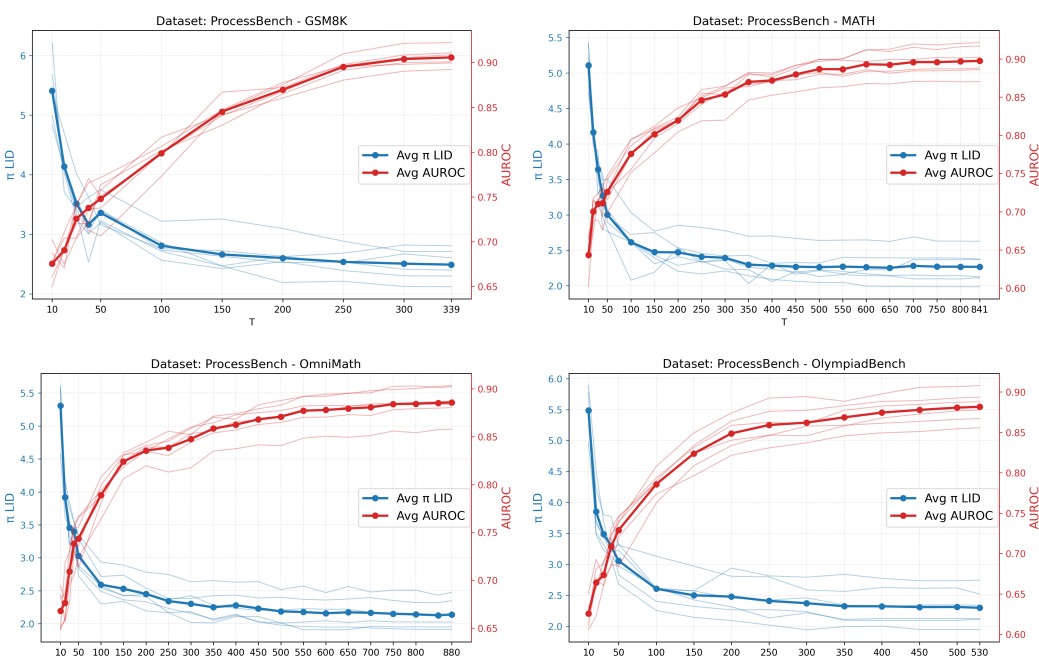

Figure 5: Neighborhood size $T$ vs. base-model LID and ProcessLID AUROC. For each Process-Bench sub-dataset, we vary $T$ (the number of nearest neighbors used to compute LID) and plot the average base-model LID $m_\pi(E)$ (blue) and the ProcessLID step-level performance (red). Semi-transparent lines show per–base-model curves; solid lines are averages over all six base models.

hood is large enough to reliably estimate LID, further increases in $T$ yield diminishing returns in ProcessLID's performance.

**Practical guidance** In practice, we recommend choosing the smallest $T$ that places $m_\pi(E)$ in its plateau regime, i.e., the smallest $T$ beyond which $m_\pi(E)$ changes negligibly with additional neighbors. This choice preserves performance while controlling compute, which is especially important on larger datasets where nearest-neighbor search dominates runtime. Empirically, the following values were near-optimal in our experiment setup: $T$=300 for GSM8K, $T$=600 for MATH, $T$=700 for OmniMath, and $T$=400 for OlympiadBench.

## A.5 EMBEDDING EXTRACTION AND INFERENCE PROCEDURE DETAILS

Given a question $Q$ and a partial reasoning trajectory $S_{1:j}$ produced by an LLM $\pi$, we format them using the conversation template in Table 8. The formatted conversation is then fed to $\pi$ in two settings: (i) *Embedding Extraction* We extract the embeddings at the last token of the formatted conversation; or (ii) *Rollout* We continue generation using $\pi$ to complete $S_{1:j}$ with the maximum total token length set to 2800.

Using a single template for both settings standardizes inputs across models and experiments.

| | |
|---|---|
| **System:** | {System prompt} |
| **User:** | Now here is a new question for you to solve: {Question $Q$} |
| **Assistant:** | {Partial trajectory $S_{1:j}$} |

Table 8: Conversation template used for embedding extraction and rollout. Content of the system prompt is provided in table 9

```
You are a helpful assistant that solves math problems by reasoning
step-by-step.

Begin every response with '<think>' to initiate your reasoning process.

Break each problem into clear, logical steps that explain your thought
process before arriving at the final solution:

- Avoid trivial one-line computations:  group related operations into a
coherent step with context.

- Do **not** split a single logical inference or calculation across
multiple steps, nor solve the entire problem in one step.

Finally, you must present your final answer at the end enclosed in
$\boxed{...}$.

Below are a few worked examples of math problems.

Carefully observe the structure and flow of the reasoning steps in each
example, paying close attention to how each step builds on the previous
one.

Use these examples as templates for your own step-by-step solutions.

{Example question and solution}
```

Table 9: System prompt used for embedding extraction and rollout.

## A.6 COMPUTING LENGTH-NORMALIZED ENTROPY

For a model vocabulary of size $|\mathcal{V}|$ and token-level predictive probabilities $p \in \Delta^{|\mathcal{V}|-1}$ (from a softmax over logits), we approximate the token entropy by retaining only the top-$K$ probabilities and aggregating the remaining tail mass in two complementary ways. We use $K = 64$ in all experiments.

**Top-$K$ split and tail mass** Let $S_K \subset \mathcal{V}$ be the indices of the $K$ largest probabilities and write

$$p_{\text{top}}(v) = \begin{cases} p(v), & v \in S_K, \\ 0, & \text{otherwise,} \end{cases} \qquad m_{\text{tail}} = \sum_{v \notin S_K} p(v) = 1 - \sum_{v \in S_K} p(v).$$

We never renormalize $p_{\text{top}}$; the tail is modeled explicitly so that the full distribution still sums to 1.

**Two tail models** We form two approximations for the unknown tail distribution:

1. Uniform tail over the $|\mathcal{V}| - K$ unseen types:

$$\tilde{p}_{\text{uni}}(v) = \begin{cases} p(v), & v \in S_K, \\ \dfrac{m_{\text{tail}}}{|\mathcal{V}| - K}, & v \notin S_K. \end{cases}$$

Its entropy is

$$H_{\text{uni}} = -\sum_{v \in S_K} p(v) \log p(v) - m_{\text{tail}} \log\left(\frac{m_{\text{tail}}}{|\mathcal{V}| - K}\right).$$

2. Collapsed tail into a single "other" symbol $\langle\text{other}\rangle$:

$$\tilde{p}_{\text{col}}(v) = \begin{cases} p(v), & v \in S_K, \\ m_{\text{tail}}, & v = \langle\text{other}\rangle, \end{cases}$$

with entropy

$$H_{\text{col}} = -\sum_{v \in S_K} p(v) \log p(v) - m_{\text{tail}} \log m_{\text{tail}}.$$

**Token entropy and step LN entropy** We average the two token-level entropies to hedge between the two tail assumptions:

$$H_{\text{token}} = \tfrac{1}{2}\left(H_{\text{uni}} + H_{\text{col}}\right).$$

For a reasoning step consisting of a sequence of tokens $(t_1, \ldots, t_L)$, the length-normalized entropy is the simple average:

$$\text{LN-Entropy} = \frac{1}{L}\sum_{\ell=1}^{L} H_{\text{token}}(t_\ell).$$

## A.7 ADDITIONAL STEP-LEVEL BENCHMARKING RESULTS

Table 10 reports the step-level AUROC computed by combining scores from all steps and evaluating one AUROC (i.e., without first computing AUROC per step and then averaging).

| Method / Model | LLAMA-3.1-8B | DS-QW-1.5B | DS-QW-7B | DS-LLAMA-8B | DS-QW-14B | DS-QW-32B | *Average* |
|---|---|---|---|---|---|---|---|
| | | | ProcessBench GSM8K | | | | |
| Random | - | - | - | - | - | - | 49.90 |
| CoT Entropy | - | - | - | - | - | - | 83.37 |
| LN Entropy | 68.14 | 70.02 | 73.55 | 68.40 | 75.13 | 72.52 | 71.29 |
| Chain of Embeddings | 52.88 | 66.36 | 69.50 | 61.75 | 59.15 | 68.59 | 63.04 |
| Vanilla LID | 74.06 | 76.35 | 71.68 | 68.20 | 77.41 | 75.46 | 73.86 |
| ProcessLID (Ours) | **87.82** | **87.59** | **86.97** | **86.81** | **89.81** | **88.40** | **87.90** |
| | | | ProcessBench MATH | | | | |
| Random | - | - | - | - | - | - | 49.50 |
| CoT Entropy | - | - | - | - | - | - | 79.67 |
| LN Entropy | 65.58 | 70.24 | 70.21 | 67.77 | 71.51 | 70.55 | 69.31 |
| Chain of Embeddings | 59.66 | 69.72 | 72.40 | 73.30 | 71.56 | 76.53 | 70.53 |
| Vanilla LID | 73.91 | 75.64 | 73.40 | 69.49 | 76.46 | 72.30 | 73.53 |
| ProcessLID (Ours) | **83.11** | **82.36** | **84.11** | **80.99** | **88.08** | **87.72** | **84.40** |
| | | | ProcessBench OmniMath | | | | |
| Random | - | - | - | - | - | - | 51.29 |
| CoT Entropy | - | - | - | - | - | - | 75.19 |
| LN Entropy | 66.44 | 70.05 | 70.59 | 68.92 | 71.84 | 71.39 | 69.87 |
| Chain of Embeddings | 55.71 | 69.47 | 73.75 | 73.27 | 70.30 | 75.45 | 69.66 |
| Vanilla LID | 70.88 | 74.55 | 73.48 | 67.76 | 74.23 | 75.82 | 72.79 |
| ProcessLID (Ours) | **82.26** | **82.27** | **84.07** | **79.54** | **85.75** | **86.31** | **83.36** |
| | | | ProcessBench OlympiadBench | | | | |
| Random | - | - | - | - | - | - | 50.36 |
| CoT Entropy | - | - | - | - | - | - | 72.94 |
| LN Entropy | 66.25 | 70.53 | 70.78 | 68.34 | 72.45 | 72.32 | 70.11 |
| Chain of Embeddings | 60.79 | 71.62 | 74.77 | 73.64 | 71.85 | 77.53 | 71.7 |
| Vanilla LID | 73.15 | 72.57 | 69.80 | 65.88 | 70.59 | 71.84 | 70.64 |
| ProcessLID (Ours) | **83.75** | **85.08** | **83.99** | **81.36** | **86.61** | **86.43** | **84.54** |

Table 10: Main results of step-level AUROC by combining scores from all steps and evaluating one AUROC for each sub-dataset for ProcessBench. Best results are in **bold**.

## A.8 ADDITIONAL PROVER MODEL ABLATION RESULTS

Figure 6 presents the prover model ablation results described in 6.2 for the four sub-datasets of ProcessBench.

## A.9 FORMAL DEFINITION ON RESIDUAL AND ALIGNMENT

In the main text (Section 6.3), we introduced *Alignment* at a high level as the correctness information contributed by the prover signal that is not already captured by the base signal. Here we provide the formal definitions.

**Signals.** For each labeled reasoning step $s_{i,j}$, we define:

• The base model signal: the Step LID Delta

$$\Delta m_\pi(E_{\pi,j}^{(i)}) := m_\pi(E_{\pi,j}^{(i)}) - m_\pi(E_{\pi,j-1}^{(i)}).$$

• The prover signal: the LID Embedding Delta

$$m_\mu(\Delta E_{\mu,j}^{(i)}) := m_\mu(E_{\mu,j}^{(i)} - E_{\mu,j-1}^{(i)}).$$

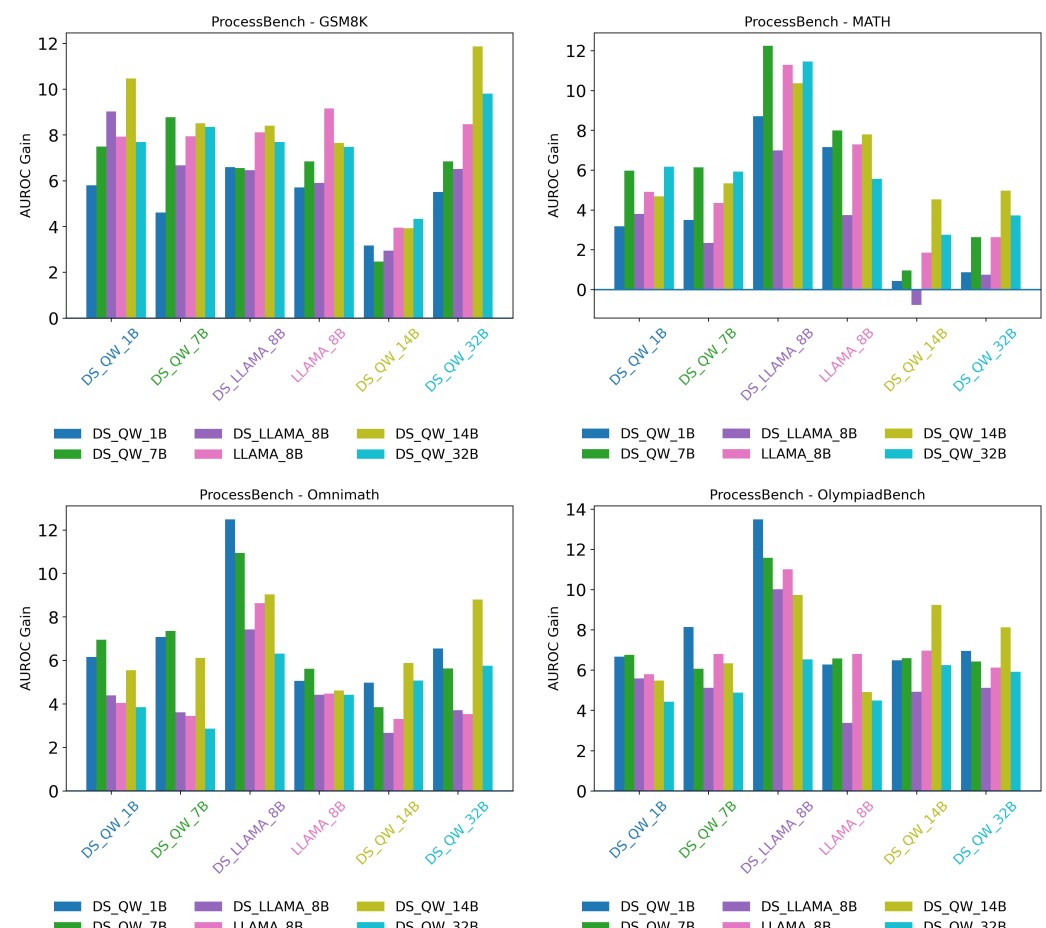

Figure 6: AUROC improvement from adding a prover $\mu$, relative to the *w/o Prover* formulation (Section 6.1). Bars are grouped by base model $\pi$ (x-axis). Within each base model, bars correspond to candidate provers (legend), ordered by increasing parameter size.

- The ProcessLID signal:

$$R_j^{(i)} = \Delta m_\pi(E_{\pi,j}^{(i)}) + \alpha \, m_\mu(\Delta E_{\mu,j}^{(i)}).$$

Each step has a ground-truth correctness label $\hat{y}_{i,j} \in \{0, 1\}$.

**Residualizing the prover signal.** To quantify how much information the prover contributes beyond the base model, we compute a nonparametric residual. Specifically, we estimate the conditional mean

$$\hat{\mathbb{E}}\big[m_\mu(\Delta E_{\mu,j}^{(i)}) \mid \Delta m_\pi(E_{\pi,j}^{(i)})\big]$$

using kernel regression, and define the residual

$$r_{i,j} = m_\mu(\Delta E_{\mu,j}^{(i)}) - \hat{\mathbb{E}}\big[m_\mu(\Delta E_{\mu,j}^{(i)}) \mid \Delta m_\pi(E_{\pi,j}^{(i)})\big].$$

This residual removes (possibly nonlinear) dependence of the prover's signal on the base signal, leaving the incremental component unique to the prover. We provide empirical evidence supporting the validity of this decomposition in Appendix A.10.

**Alignment.** We formally define Alignment as the mutual information between the residual and the ground-truth step labels:

$$\text{Align} := I(\mathbf{r}; \hat{\mathbf{Y}}).$$

Here $\mathbf{r}$ is the collection of residuals across all steps and $\hat{\mathbf{Y}}$ the corresponding labels. Intuitively, Alignment measures how much *new* label-relevant information the prover contributes beyond what is already present in the base model's signal. Larger values correspond to stronger complementary information from the prover.

## A.10 ANALYSIS ON THE RESIDUAL DECOMPOSITION OF PROCESSLID

| Method / Base Model | LLAMA-3.1-8B | DS-QW-1.5B | DS-QW-7B | DS-LLAMA-8B | DS-QW-14B | DS-QW-32B | Average |
|---|---|---|---|---|---|---|---|
| | | | ProcessBench | | | | |
| Cond.-Mean Only | 80.12 | 73.95 | 75.84 | 65.85 | 77.60 | 74.76 | 74.69 |
| w/o Prover | 80.44 | 74.70 | 76.09 | 66.43 | 78.20 | 75.45 | 75.22 |
| Residual Only | 87.18 | **87.25** | 86.80 | **84.52** | 88.92 | 87.20 | 86.98 |
| ProcessLID (Original) | **87.21** | 86.65 | **86.88** | 84.17 | **89.13** | **88.05** | **87.01** |

Table 11: Step-level AUROC on ProcessBench comparing the full ProcessLID to ablations that isolate the conditional-mean or residual component of the prover signal. The prover is fixed to DS-QW-14B for all base models and formulations. Best results are in **bold**.

To verify the validity of the decomposition

$$m_\mu(\Delta E) \;=\; \underbrace{\mathbb{E}[m_\mu(\Delta E) \mid \Delta m_\pi(E)]}_{\text{conditional mean}} + \underbrace{r}_{\text{residual}},$$

we compare ProcessLID $\big(\Delta m_\pi(E_{\pi,j}) + \alpha\, m_\mu(\Delta E_{\mu,j})\big)$ against three targeted variants on Process-Bench using the step-level experiment described in Section 5.1:

1. *Cond.-Mean Only* replaces the prover term with the estimated conditional mean, removing any residual contribution:

$$\Delta m_\pi(E_{\pi,j}) \;+\; \alpha\, \widehat{\mathbb{E}}[m_\mu(\Delta E_{\mu,j}) \mid \Delta m_\pi(E_{\pi,j})]\,.$$

2. *Residual Only* keeps only the residual component $r_j$ added to the base signal:

$$\Delta m_\pi(E_{\pi,j}) \;+\; \alpha\, r_j.$$

3. *w/o Prover* removes the prover entirely and uses the base signal alone:

$$\Delta m_\pi(E_{\pi,j}).$$

Including *w/o Prover* provides a more informative baseline: it isolates the value of any prover-derived information and clarifies whether improvements stem from the conditional-mean, the residual, or simply from the base model's signal.

Table 11 shows that *Cond.-Mean Only* suffers a substantial drop (7–18 AUROC below ProcessLID) and performs on par with *w/o Prover*. This indicates that the conditional-mean component of the prover's signal adds little label-relevant information beyond what is already captured by the base signal $\Delta m_\pi(E)$. In contrast, *Residual Only* matches the full ProcessLID and occasionally slightly exceeds it, demonstrating that the incremental, base-model orthogonal component $r$ is the part of the prover signal that carries most of the added discriminative power. These results empirically validate the decomposition and justify focusing on the residual as the primary source of complementary information.

## A.11 ADDITIONAL ALIGNMENT-SEPARABILITY ANALYSIS RESULTS

Figure 7 presents the alignment-separability analysis results for the additional four base models described in section 6.3.

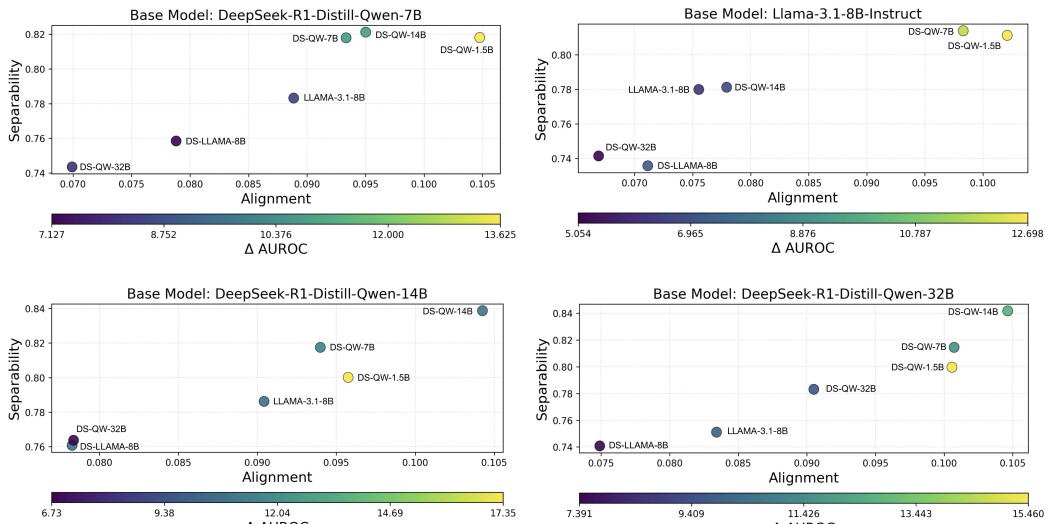

Figure 7: AUROC gain for candidate prover models $\mu$ as a function of Alignment (x-axis) and Separability (y-axis), with the base model fixed. Each point is labeled by $\mu$; point color encodes the AUROC gain contributed by that prover. The accompanying color bar provides the mapping from color to the numeric AUROC gain (darker = smaller gain, brighter = larger gain)

