# OpenReview forum: "ProcessLID: Step-wise Internal Reward in LLM Reasoning via Local Intrinsic Dimension"
_ICLR.cc/2026/Conference — Submitted to ICLR 2026_

### Official Review · Reviewer_NmXq · 2025-10-17

**Soundness:** 3
**Presentation:** 2
**Contribution:** 3
**Rating:** 6
**Confidence:** 3

**Summary:**

This paper introduces ProcessLID, a novel and training-free method for assessing the step-level correctness of LLM reasoning in mathematical problem-solving. Grounded in the concept of LID, the method innovates by using a step LID delta to isolate the marginal contribution of each reasoning step and incorporates a distinct 'prover' model to provide a complementary correctness signal from a different representation geometry. The final signal, an aggregation of these two components across multiple model layers, is shown to achieve sota performance in offline correctness classification and remains highly competitive with strong, trained reward models in inference-time answer selection tasks.

**Strengths:**

* Novelty: The paper proposes a highly innovative, training-free approach to a critical problem. The method is well-grounded in a concept (LID), and its core innovations—the step LID delta for isolating step-correctness and the use of a prover model for a complementary signal.

* Empirical performance and theoretical robustness: ProcessLID achieves sota results on offline benchmarks and demonstrates remarkable competitiveness against a fully fine-tuned reward model in inference-time settings. Besiddes, The paper excels in its deep analysis of why the proposed components work.

* Practicality: Eliminating the need for expensive data collection(MCTS) and model training, ProcessLID presents a highly practical and computationally efficient alternative to traditional scalar reward-based prms.

**Weaknesses:**

* Inference Cost Analysis: While ProcessLID is training-free, it introduces specific inference-time costs, namely the nearest-neighbor search for LID calculation and an additional forward pass for the prover model. The paper would be strengthened by a more explicit analysis of these computational overheads in comparison to a standard PRM.

* Generalization: How does the performance of ProcessLID degrade if the reference dataset for the nearest-neighbor search is out-of-domain. A discussion on the method's sensitivity should be included.

* Clear role of prover: Did you experiment with using a much weaker prover (e.g., a 1.5B model as a prover for a 32B base model)? Does a significant capability gap negatively impact the prover's alignment?

* Stronger baselines should be considered, such as Qwen3.

**Questions:**

* For the inference-time experiments, could you elaborate on the decision to use fixed-length 100-token steps? How sensitive is the performance to this choice, and have you explored using semantic or logical step delimiters instead? (most commonly, \n\n)

---

> ### Author Response · Authors · 2025-11-27
> **Addressing the inference cost analysis**
>
> We thank the reviewer for raising the concern about the inference-time overhead introduced by the nearest-neighbor search and the additional forward pass through the prover model.
>
> We respectfully direct the reviewer to our public comment *Runtime Analysis* in this thread, where we provide a detailed breakdown of the runtime for ProcessLID and all baselines, including how each component contributes to the total inference cost.
>
> Below we summarize the key observations most relevant to this concern:
>
> 1. **Nearest-neighbor search is negligible relative to forward passes.**
>    In our experiments, we use 500 reference datapoints, which we found (Appendix A.4) to be a good trade-off between performance and efficiency. Under this setup, the runtime overhead of nearest-neighbor search in ProcessLID is negligible. For all internal-representation–based methods (including ProcessLID), the dominant cost is the forward pass through the generator (and, for ProcessLID, the prover) to obtain embeddings. Thus, the main runtime concern lies in the prover forward pass rather than in the nearest-neighbor lookup itself.
>
> 2. **Prover overhead can be substantially reduced with small, aligned models.**
>    Although the additional forward pass through the prover model dominates ProcessLID’s extra cost, we show in Sections 6.2 and 6.3 that this overhead can be significantly reduced with minimal performance loss by using a smaller but aligned prover. In particular, the *ProcessLID Light* variant (using DS-QW-1.5B instead of DS-QW-14B as the prover model), presented in the public comment *Runtime Analysis*, retains **98%** of the original ProcessLID’s performance while requiring only **41.5%** of its runtime (and is just **17%** slower than vanilla LID), and it still outperforms all other baselines.
>
> 3. **Overall runtime is comparable to other strong baselines.**
>    Our runtime study shows that ProcessLID has the same order-of-magnitude runtime as other strong baselines: LN Entropy is substantially faster (≈7.7×), while CoT Entropy is substantially slower (≈4.2×). This indicates that, in practice, ProcessLID’s inference cost is modest rather than prohibitive.
>
> Taken together, these findings suggest that although we do use hundreds of reference problems, the associated nearest-neighbor search cost is negligible compared to model forward passes, and the additional prover forward pass can be significantly reduced via smaller, aligned provers with minimal performance compromise.

---

> ### Author Response · Authors · 2025-11-27
> **Addressing the clear role of prover**
>
> We thank the reviewer for raising this question about whether a weaker prover, or a large capability gap between the base and prover, negatively impacts ProcessLID’s performance.
>
> We highlight Section 6.2 contains detailed ablation over all base–prover combinations. Given our six base models, we evaluate ProcessLID on ProcessBench for all 36 possible (base model, prover model) pairs and report their step-level AUROC gains over ProcessLID formulated with only the base model term ($\Delta m_{\pi}(E_{\pi,j})$). These results are visualized in Figure 2, and we additionally provide a tabular summary here:
>
> |                               |   DS-QW-1.5B |   DS-QW-7B |   LLAMA-3.1-8B |   DS-LLAMA-8B |   DS-QW-14B |   DS-QW-32B |   Overall |
> |:------------------------------|--------------------------------:|------------------------------:|------------------------:|-------------------------------:|-------------------------------:|-------------------------------:|----------:|
> | DS-QW-1.5B |                            5.45 |                          5.83 |                    6.04 |                          10.32 |                           3.76 |                           4.96 |      6.06 |
> | DS-QW-7B   |                            6.79 |                          7.08 |                    6.76 |                          10.33 |                           3.47 |                           5.38 |      6.63 |
> | LLAMA-3.1-8B         |                            5.67 |                          5.63 |                    6.93 |                           9.76 |                           4.02 |                           5.19 |      6.2  |
> | DS-LLAMA-8B  |                            5.7  |                          4.44 |                    4.36 |                           7.72 |                           2.44 |                           4.02 |      4.78 |
> | DS-QW-14B  |                            6.54 |                          6.57 |                    6.24 |                           9.38 |                           5.89 |                           8.44 |      7.18 |
> | DS-QW-32B  |                            5.53 |                          5.5  |                    5.49 |                           7.99 |                           4.6  |                           6.3  |      5.9  |
>
> In this table, row names indicate the base model and column names indicate the prover model, and each cell shows the **AUROC gain** of that base–prover pair over ProcessLID with only the base model term. The rightmost “Overall” column aggregates the gains for each prover by averaging over all base models. We summarize the key findings here:
>
> - Even very small prover models provide substantial gains. The smallest prover, **DS-QW-1.5B**, yields an average improvement of **6.06 AUROC** across all base models, which is only **1.12 AUROC** below the best prover, **DS-QW-14B** (average gain **7.18 AUROC**).
> - The second-to-last smallest prover **DS-QW-7B** achieves an average gain of **6.63 AUROC**, trailing the best prover DS-QW-14B by only **0.55 AUROC**.
> - Crucially for this comment, for the **strongest base model DS-QW-32B**, using the **smallest prover DS-QW-1.5B** still yields a gain of **4.96 AUROC** over ProcessLID with only the base model term. This indicates that even with a significant capability gap (32B base vs. 1.5B prover), ProcessLID continues to improve performance rather than being negatively affected.
>
> Overall, these results show that (i) small and less capable provers remain highly effective, and (ii) we do not observe performance degradation or misalignment when there is a large capability gap between the base and prover. Instead, ProcessLID consistently benefits from the prover’s **distinct representation geometry**, rather than its nominal size or capacity, across a wide range of base–prover configurations (analyzed more formally in Scetion 6.2). This gives it a competitive advantage over methods that rely directly on an external judge LLM’s raw problem-solving ability. For example, CoT Entropy shows a consistent degradation in step-level AUROC on ProcessBench, dropping from 88.83 on the easiest (grade-school) sub-dataset to 77.78 on the hardest Olympiad-level sub-dataset as task difficulty increases. In contrast, even weak but aligned provers (e.g., 1.5B) continue to provide substantial gains for ProcessLID across all difficulty levels, indicating that our improvements are driven primarily by geometric complementarity rather than the prover’s scale.

---

> ### Author Response · Authors · 2025-11-27
> **Addressing step segmentation choice in inference time experiment**
>
> We thank the reviewer for the insightful question regarding our step segmentation choice in the inference-time experiments.
>
> We agree that segmenting rollouts into fixed 100-token steps is not well aligned with modern reasoning paradigms, where steps are naturally of variable length and often organized into semantically coherent blocks. In response to this concern, we have **updated our inference-time setup** to evaluate ProcessLID on steps segmented by a semantic delimiter `\n\n`, which better approximates natural reasoning units (e.g., paragraphs or lines containing self-contained logical moves).
>
> The updated inference-time results are reported in Table 3 (Section 5.2). The key observations are:
>
> - On *Oracle@N (Pruned-5)* and *Weighted Majority Vote*, ProcessLID now **outperforms** the reward model for both generator models by an average of **2.00** and **0.33** AUROC, respectively.
> - On *Best of N*, ProcessLID trails the reward model by only **1.82** AUROC on average (versus a **5.66**-point gap in the original fixed-length setup).
>
> We emphasize that the reward model is a high-capacity 8B sequence scorer trained with substantial human preference data and compute, whereas ProcessLID is fully training-free and uses only internal representations. In this context, surpassing the reward model on two metrics and remaining highly competitive on *Best of N* under a semantically segmented setup provides stronger evidence that ProcessLID has meaningful end-to-end impact on problem-solving performance.
>
> To directly address sensitivity to the segmentation scheme, we compare ProcessLID’s inference time performance under the original 100-token segmentation (“Old”) and the revised `\n\n`-based segmentation (“New”):
>
> |                           | LLAMA-3.1-8B         | DS-LLAMA-8B          | Average        |
> |---------------------------|----------------------|----------------------|----------------|
> | **Oracle @ N Pruned 5**   |                      |                      |                |
> | ProcessLID (Old)          | $$74.89_{0.36}$$     | $$81.78_{0.36}$$     | $$78.34$$      |
> | ProcessLID (New)          | $$78.00_{3.20}$$     | $$84.00_{1.85}$$     | $$81.00$$      |
> | **Weighted Majority Vote**|                      |                      |                |
> | ProcessLID (Old)          | $$62.22_{1.28}$$     | $$80.00_{0.62}$$     | $$71.11$$      |
> | ProcessLID (New)          | $$65.33_{3.20}$$     | $$79.33_{1.85}$$     | $$72.33$$      |
> | **Best of N**             |                      |                      |                |
> | ProcessLID (Old)          | $$54.67_{1.23}$$     | $$69.56_{0.36}$$     | $$62.12$$      |
> | ProcessLID (New)          | $$57.33_{2.13}$$     | $$74.59_{2.97}$$     | $$65.96$$      |
>
> Switching from fixed 100-token segments to `\n\n`-delimited segments yields consistent improvements across all three metrics: **+2.66** (Oracle@N Pruned-5), **+1.22** (Weighted Majority Vote), and **+3.84** (Best of N). This behavior is consistent with our design intuition: **Step LID Delta** and **LID Embedding Delta** are meant to capture the *marginal correctness contribution* of each reasoning step, which is better reflected when steps are semantically coherent rather than arbitrary token slices.

---

### Official Review · Reviewer_NGTy · 2025-10-28

**Soundness:** 2
**Presentation:** 2
**Contribution:** 2
**Rating:** 2
**Confidence:** 4

**Summary:**

This paper introduces ProcessLID, a  training-free method for generating step-level correctness signals for mathematical reasoning in LLMs. The problem addressed is the high cost and limited granularity of existing methods for evaluating reasoning steps. ProcessLID is based on the concept of Local Intrinsic Dimension (LID), which quantifies the complexity of a model's representation manifold. The core innovations are: 1) using a "Step LID Delta" to isolate the correctness of a single reasoning step by comparing LID before and after the step, 2) employing a separate "prover" model to provide a complementary correctness signal from a distinct representation geometry, and 3) aggregating these signals across multiple model layers. Experiments across six models and four math benchmarks show that ProcessLID achieves state-of-the-art performance in offline correctness classification and is competitive in inference-time answer selection scenarios.

**Strengths:**

- This paper using a Step LID Delta to isolate the marginal contribution of a single reasoning step, which is a clever way to adapt LID, which is typically a static measure, to a dynamic, step-by-step process.
- The evaluation spans six different generator models and four mathematical reasoning benchmarks of increasing difficulty.

**Weaknesses:**

**Questionable Motivation for LID in Mathematical Reasoning**:

*   The central hypothesis is that incorrect reasoning steps are sampled from a more complex, "fabricated model distribution" and thus have a higher LID, while correct steps narrow the solution space and have a lower LID. This is analogous to out-of-distribution detection in QA.
*   However, mathematical reasoning is fundamentally different. Incorrect steps (e.g., calculation errors) are common even in human reasoning and do not necessarily represent a major deviation into an "unnatural" manifold. A model can often recover from a minor error in a subsequent step.
*   Conversely, a correct reasoning path can involve exploring multiple branches or introducing complex new concepts, which might not necessarily "compress the reachable representation states" as the paper claims. This untested assumption about the relationship between correctness and representation complexity in math reasoning weakens the method's conceptual foundation.

**Practicality Concerns and Unquantified Overhead**: While the method is training-free, it introduces dependencies and computational costs that affect its practical application.

*   The method's performance is heavily reliant on the choice of the prover model μ, as shown in the ablation study and prover analysis. The paper provides a post-hoc analysis of what makes a good prover but offers little practical guidance on how to select one *a priori* without extensive experimentation.
*   ProcessLID requires a full forward pass through the prover model for every reasoning step. This adds a significant computational overhead compared to baselines that only use the base model's representations. The paper claims competitiveness in "inference-time settings" but does not provide any measurements  to quantify this overhead.
*   The computation of LID relies on a nearest-neighbor search over a set of CoTs from the same dataset. This introduces a data dependency, requiring a sufficiently large and in-domain corpus of reasoning trajectories to be available at inference time, which may not always be practical. The computational cost of this search is also not discussed.

**Limited Scope of Inference-Time Experiments**:

*   The decision to segment rollouts into fixed-length steps of 100 tokens is not well-justified. Natural reasoning steps are variable in length, and this arbitrary segmentation might not align with the logical structure of the reasoning process. The sensitivity of the results to this hyperparameter is not explored.
*   The evaluation is restricted to reranking-based answer selection (Oracle@N, BoN, etc.). It would be valuable to explore or at least discuss how ProcessLID could be integrated more deeply into the generation process, for example, by guiding a beam search or speculative decoding to prune incorrect reasoning paths early.
*   While competitive, ProcessLID shows a noticeable performance gap compared to the supervised Reward Model on the BoN metric, which is often critical for improving final answer accuracy. This limitation could be more explicitly acknowledged and discussed.

**Questions:**

see weakness and:

**Clarification on LID for Embedding Deltas**: Regarding the definition of the LID Embedding Delta in Section 4.3 (line 157), could the work provide a more precise mathematical formulation? Specifically, is the nearest neighbor search for computing `mµ(ΔΕμ,j)` performed over a pre-computed set of difference vectors ` {E(t)μ,j – E(t)μ,j−1} ` from the reference dataset? If so, what is the geometric intuition for measuring the intrinsic dimension of a manifold of such difference vectors?

---

> ### Author Response · Authors · 2025-11-27
> **Addressing the lack of practical guidance and high sensitivity on prover selection**
>
> We thank the reviewer for raising these important concerns about the practical guidance for choosing a prover model and the method sensitivity.
>
> We agree that, in the initial presentation, our discussion of prover choice focused more on post-hoc analysis than on actionable guidance. However, we would like to emphasize that the prover ablation results actually show that ProcessLID is *robust* to the choice of prover, rather than highly sensitive.
>
> For clarity, we restate the tabular form of the prover ablation (originally summarized in Figure 2):
>
> |                               |   DS-QW-1.5B |   DS-QW-7B |   LLAMA-3.1-8B |   DS-LLAMA-8B |   DS-QW-14B |   DS-QW-32B |   Overall |
> |:------------------------------|-------------:|-----------:|---------------:|--------------:|------------:|------------:|----------:|
> | DS-QW-1.5B                    |        5.45  |      5.83  |          6.04  |        10.32  |        3.76 |        4.96 |     6.06  |
> | DS-QW-7B                      |        6.79  |      7.08  |          6.76  |        10.33  |        3.47 |        5.38 |     6.63  |
> | LLAMA-3.1-8B                  |        5.67  |      5.63  |          6.93  |         9.76  |        4.02 |        5.19 |     6.20  |
> | DS-LLAMA-8B                   |        5.70  |      4.44  |          4.36  |         7.72  |        2.44 |        4.02 |     4.78  |
> | DS-QW-14B                     |        6.54  |      6.57  |          6.24  |         9.38  |        5.89 |        8.44 |     7.18  |
> | DS-QW-32B                     |        5.53  |      5.50  |          5.49  |         7.99  |        4.60 |        6.30 |     5.90  |
> | **Overall**                   |   5.95±0.52  |  5.84±0.84 |     5.97±0.86  |    9.25±1.04  |   4.03±1.06 |   5.71±1.39 |  6.12±0.73 |
>
> Here, each entry reports the *AUROC gain* of a given base–prover pair over a ProcessLID variant that uses only the base model term $\Delta m_{\pi}(E_{\pi,j})$, so it isolates the contribution of the prover component. Row names index the base model, column names index the prover model, and the rightmost “Overall” column averages over all bases. The last row reports the mean and standard deviation of the AUROC gains across all provers for each base model.
>
> Two key observations address the sensitivity concern:
>
> - For *every* prover model, the average gain over the base-only variant is substantial (from 4.78 to 7.18 AUROC), showing that a wide range of provers meaningfully improve performance.
> - The standard deviations of these gains per base model are small (between 0.52 and 1.39), indicating that performance does not fluctuate wildly with different prover choices.
>
> Thus, ProcessLID is not brittle with respect to prover choice; instead, many reasonable provers — including relatively small ones — yield strong and stable gains. In practice, this means that one does not need to finely tune the prover model: a general-purpose LLM with reasonable math/reasoning capability is already expected to provide substantial improvement over the base model term-only ProcessLID variant.
>
> Our post-hoc analysis of prover effectiveness in Section 6.3 should be interpreted in this context. It is built on the observation that a wide range of prover models already contribute substantial performance gains, and focuses on understanding how and why the prover term helps so that practitioners can maximize these gains (e.g., by favoring provers with well-aligned representation geometry), rather than on mitigating a fragile dependence on a single, uniquely “correct” prover choice.
>
> These results suggest that ProcessLID’s performance is robust rather than highly sensitive to the specific choice of prover—so in practice, a reasonably capable general-purpose LLM already yields substantial gains over the base model term-only ProcessLID variant.

---

> ### Author Response · Authors · 2025-11-27
> **Addressing the computational overhead of nearest neighbor search and prover model forward pass in ProcessLID**
>
> We thank the reviewer for raising these important concerns about (i) the inference-time overhead of the prover forward pass, and (ii) the practicality and cost of the nearest-neighbor search over reference CoTs.
>
> Regarding the inference-time cost of the prover forward pass and the nearest-neighbor search, we respectfully refer the reviewer to our public comment *Runtime Analysis*, where we provide a detailed breakdown of runtime for ProcessLID and all baselines. Here we summarize the points most relevant to this comment:
>
> 1. **Nearest-neighbor search cost is negligible relative to model forward passes.**
>    In our experiments, we use 500 reference datapoints, which we found (Appendix A.4) to be a good trade-off between performance and efficiency. With this configuration, the FAISS-based nearest-neighbor search contributes only a small fraction of the total runtime. For all internal-representation–based methods (including ProcessLID), the dominant cost is the forward pass through the generator (and, for ProcessLID, the prover) to obtain embeddings. In practice, the main overhead comes from the prover forward pass, not from nearest-neighbor queries.
>
> 2. **Prover overhead can be substantially reduced with small, aligned models.**
>    While the additional forward pass through the prover does increase inference-time cost, we show in Sections 6.2 and 6.3 that this overhead can be significantly reduced with minimal performance loss by using a smaller but aligned prover. Concretely, the *ProcessLID Light* variant (which uses DS-QW-1.5B instead of DS-QW-14B as the prover) retains **98%** of the original ProcessLID’s performance, requires only **41.5%** of its runtime, and is just **17%** slower than vanilla LID, while still outperforming all non-ProcessLID baselines. This demonstrates that the extra cost is controllable in practice.
>
> 3. **Overall runtime remains comparable to other strong baselines.**
>    Our runtime study shows that ProcessLID has the same order-of-magnitude runtime as other strong baselines: LN Entropy is substantially faster (≈7.7×), whereas CoT Entropy is substantially slower (≈4.2×). This suggests that ProcessLID’s inference-time overhead is modest rather than prohibitive, especially when combined with smaller provers as above.
>
> These results suggest that the main inference-time overhead arises from a single additional forward pass through the prover model, which can be substantially reduced by using smaller yet aligned provers, while incurring only minimal compromise in performance.

---

> ### Author Response · Authors · 2025-11-27
> **Addressing the limited scope of inference time experiments**
>
> We thank the reviewer for the thoughtful comments on (i) the choice of fixed-length step segmentation, (ii) the limited scope of our inference-time experiments, and (iii) the comparison to the supervised reward model on the Best-of-N metric.
>
> **Step segmentation and sensitivity to step definition.**
> We agree that fixed-length 100-token segmentation is not well aligned with modern reasoning paradigms, where natural reasoning steps have variable length. In response, we have revised our inference-time setup to better reflect this. Instead of segmenting rollouts into fixed-length blocks of 100 tokens, we now segment steps using the delimiter `\n\n`, which yields semantically more self-contained units (e.g., roughly coherent reasoning chunks). This step definition is better matched to our Step LID Delta and LID Embedding Delta formulations, which are designed to evaluate the correctness of individual reasoning moves.
>
> Under this revised segmentation, we also include vanilla LID as a baseline evaluated under exactly the same rollout-selection setup as ProcessLID. The updated inference-time results are reported in Table 3 (Section 5.2). In summary:
>
> - On **Oracle@N (Pruned-5)** and **Weighted Majority Vote**, ProcessLID now *outperforms* the supervised reward model for both generator models, with average gains of 2.00 AUROC and 0.33 AUROC, respectively.
> - On **Best-of-N**, ProcessLID trails the reward model by only 1.82 AUROC on average (compared to a gap of 5.66 AUROC in the earlier 100-token segmentation setup).
>
> Given that the reward model is a high-capacity 8B sequence scorer trained with substantial human preference data and task-specific supervision, whereas ProcessLID is entirely training-free and uses only internal representations, surpassing the reward model on two metrics and remaining very competitive on Best-of-N provides strong evidence that ProcessLID has substantial end-to-end impact on problem-solving performance and multi-step reasoning quality.
>
> **Beyond reranking: integration into generation (beam search, pruning, and reflective control).**
> We agree that a deeper integration of ProcessLID into the generation process (e.g., beam search, speculative decoding, or tree-style reasoning) is an important next step that goes beyond reranking. In the current work, we intentionally focus on the reranking-based setting because it offers a clean, widely used protocol that isolates the quality of the *step-level scoring function* without entangling it with decoding heuristics.
>
> That said, ProcessLID is naturally suited for more interactive integration during generation. Concretely, one can envision:
>
> - **Beam / branch pruning at semantic boundaries.**
>   At each semantic step boundary (e.g., when a `\n\n` delimiter is produced), we can compute ProcessLID scores for candidate continuations along different beams or rollouts. These scores can then be used to prune beams whose next step appears geometrically off-manifold (high ProcessLID), or to reweight beam scores so that trajectories with consistently low ProcessLID across steps are favored.
>
> - **Step-wise reflective control.**
>   In addition, ProcessLID enables an adaptive “reflect-when-necessary” mechanism. During generation, we can continuously monitor the step-wise ProcessLID score; whenever it crosses a threshold indicating that the current step is likely incorrect, the system can trigger an explicit reflection or self-correction prompt to the LLM (e.g., asking it to reassess or revise the last step before proceeding). This design both (i) steers the reasoning process back toward more plausible trajectories and (ii) preserves efficiency by invoking reflection only when the geometry-based signal indicates a high risk of error, rather than at every step.
>
> - **Integration with existing decoding strategies.**
>   These mechanisms can be combined with standard decoding strategies (beam search, speculative decoding, or tree-of-thought–style exploration) so that ProcessLID provides a principled, geometry-based criterion for which branches to continue, prune, or revise.
>
> We view these as natural extensions of our current framework. The strong inference-time performance we already observe under semantically segmented rollouts and its robustness to task difficulty when evaluated on ProcessBench suggests that ProcessLID is a promising step-level scoring primitive for such decoding-time control, and we plan to explore these directions in future work.

---

> ### Author Response · Authors · 2025-11-27
> **Clarification on LID for Embedding Deltas**
>
> We thank the reviewer for the question regarding the definition and geometric intuition of the LID Embedding Delta.
>
> When computing the LID Embedding Delta $m_{\mu}(\Delta E_{\mu,j})$, the nearest-neighbor search is indeed performed over a pre-computed set of *delta embeddings* from the reference dataset. Concretely, for each reference trajectory $t \in \lbrace1,\dots,N\rbrace$ and step index $j$, we form $\Delta E_{\mu,j}^{(t)}= E_{\mu,j}^{(t)} - E_{\mu,j-1}^{(t)}$
> and the LID Embedding Delta at test step $j$ is computed by applying the standard LID estimator to $\Delta E_{\mu,j}$ with respect to the collection $\lbrace \Delta E_{\mu,j}^{(t)}\rbrace_{t=1}^N$.
>
> The LID Embedding Delta is designed to capture the *local geometry of step-wise updates*. Each $\Delta E_{\mu,j}^{(t)}$ can be viewed as a “tangent-like” vector describing how the prover’s representation moves in response to a single reasoning step. By computing the LID of $\Delta E_{\mu,j}$ with respect to the reference set $\lbrace\Delta E_{\mu,j}^{(t)}\rbrace$, we are effectively estimating the intrinsic dimensionality of the *local manifold of plausible step-wise changes* at step $j$. Using *delta embeddings* (rather than raw embeddings) for both the query and the reference set ensures that the LID estimate is aligned with the geometry of *step-level transitions*, rather than being confounded by the absolute position of the trajectory prefix.
>
> To validate this design choice, we conducted an additional ablation in which we replaced the delta-reference set with raw embeddings. We compare our original ProcessLID formulation against a *ProcessLID w/o Delta Reference* variant. In this variant, when computing $m_{\mu}(\Delta E_{\mu,j})$, we perform nearest-neighbor search against $\lbrace E_{\mu,j}^{(t)}\rbrace_{t=1}^N$ instead of $\lbrace\Delta E_{\mu,j}^{(t)}\rbrace_{t=1}^N$, keeping all other components unchanged. The results are presented below (best step-level AUROC are in **bold**):
>
> | **Method / Model** | **LLAMA-3.1-8B** | **DS-QW-1.5B** | **DS-QW-7B** | **DS-LLAMA-8B** | **DS-QW-14B** | **DS-QW-32B** | ***Average*** |
> |---------------------------------|-----------------:|---------------:|-------------:|----------------:|--------------:|--------------:|--------------:|
> | **ProcessBench GSM8K** | | | | | | | |
> | ProcessLID (Original) | **90.88** | **89.95** | **89.22** | **90.08** | **92.22** | **91.09** | **90.57** |
> | ProcessLID w/o Delta Reference | 88.56 | 86.75 | 87.00 | 88.43 | 91.59 | 85.87 | 88.03 |
> | **ProcessBench MATH** | | | | | | | |
> | ProcessLID (Original) | **88.60** | **88.80** | **90.20** | **87.02** | **92.12** | **91.68** | **89.73** |
> | ProcessLID w/o Delta Reference | 87.76 | 86.25 | 88.95 | 84.76 | 88.00 | 87.64 | 87.23 |
> | **ProcessBench OmniMath** | | | | | | | |
> | ProcessLID (Original) | **88.35** | **88.04** | 88.72 | 85.79 | **90.21** | **90.33** | **88.57** |
> | ProcessLID w/o Delta Reference | 87.37 | 87.33 | **89.06** | **87.94** | 88.65 | 88.90 | 88.21 |
> | **ProcessBench OlympiadBench** | | | | | | | |
> | ProcessLID (Original) | **86.70** | **87.77** | **89.36** | 85.57 | **90.78** | **88.78** | **88.16** |
> | ProcessLID w/o Delta Reference | 86.32 | 85.29 | 88.19 | **85.68** | 85.61 | 86.91 | 86.33 |
>
> Across all six base models and four ProcessBench sub-datasets, the original ProcessLID formulation outperforms the *w/o Delta Reference* variant in 21 out of 24 settings. In the few cases where the variant slightly surpasses ProcessLID, the margin is always below 2.15 AUROC. On average, the original formulation improves over *ProcessLID w/o Delta Reference* by 1.8 AUROC.
>
> These results support the intuition that computing LID over *delta embeddings*—and using delta embeddings as the reference set—provides a more faithful estimate of the local intrinsic dimension of step-wise changes, and is empirically preferable to using raw embeddings for the same purpose.

---

> ### Author Response · Authors · 2025-11-27
> **Addressing the questionable motivation of LID in mathematical reasoning**
>
> We thank the reviewer for raising this concern about the validity of our central hypothesis and its applicability to mathematical reasoning.
>
> Our hypothesis is *statistical* rather than absolute: over many trajectories, **correct** steps tend to move hidden states into more constrained, training-supported regions of representation space (e.g., coherent algebraic manipulations and symbol usage), while **incorrect** steps—especially those that derail the logic or reframe the problem—more often drift into regions with more diverse, less constrained continuations. Building on this, ProcessLID does not rely on absolute LID alone, but instead uses **changes in local geometry** at each step, captured via Step LID Delta and LID Embedding Delta. These two signals provide complementary views on whether a new step tightens the manifold around a coherent intermediate state or pushes the trajectory into regions with many mutually inconsistent continuations.
>
> Empirically, the strength and robustness of our results support that this geometric view captures a meaningful correlate of correctness in math reasoning. ProcessLID consistently improves over vanilla LID and other internal-representation baselines across all four ProcessBench sub-datasets, which span difficulty from grade-school to Olympiad-level problems. In inference-time settings, ProcessLID matches or surpasses a strong supervised reward model on two of three metrics while remaining highly competitive on Best-of-N, despite being fully training-free. Moreover, these gains are stable across task difficulty and across a broad range of prover models (including relatively small provers), suggesting that the underlying geometric signal is not a fragile artifact of any single model or dataset.
>
> That said, we agree that the relationship between correctness and representation complexity in mathematical reasoning is subtle and open to alternative interpretations. Our work offers an empirically grounded but still preliminary geometric hypothesis. As future work, we plan to investigate this more directly to better understand when and why LID-based signals succeed or fail in capturing correctness.

---

### Official Review · Reviewer_kquz · 2025-10-31

**Soundness:** 3
**Presentation:** 2
**Contribution:** 2
**Rating:** 6
**Confidence:** 2

**Summary:**

ProcessLID introduces a training-free representation-based method to estimate step-level correctness in multi-step reasoning.
It refines the Local Intrinsic Dimension (LID) idea with the addition of a proxy prover model to provide step-level information on the degree to which information representation is collapsed in a reasoning chain. Authors experimented on different LLMs across multiple reasoning datasets (GSM8K, MATH, OmniMath, OlympiadBench) and demonstrates consistent performance gain. ProcessLID also performs competitively with a trained reward model during inference-time answer selection.

**Strengths:**

Having training-free internal reward is an important topic and the current work's approach of using geometric properties of representations as signal is novel and seem to be effective.

While the notations can be dense in the manuscript, the algorithm is actually relatively simple and could easily scale to more models and more reference problems.

Ablations are quite thorough in the paper and the results seem effective.

The fact that LID can be used to guide inference is quite interesting and potentially could be used as a diagnostic tool to better understand how different model families represent concepts during reasoning.

**Weaknesses:**

It is not clear to me how much the proposed problem would generalize if the reference problems are from a different domain, or in a real-life deployment when the user queries can be quite diverse. Since the underlying assumption is that the representation during reasoning train live on some kind of manifold at each given reasoning step, I'm not sure how much this assumption will hold up in more complex settings.

The paper is a bit hard to follow, the notations can be quite dense.

Since the results in appendix suggests that hundreds of reference problems will be needed for best performance, it seems relatively heavy. Perhaps we could instead look like LID when perturbing model hidden state in prefix?

**Questions:**

One thing that would help reading a lot would be to include a pseudocode description of the algorithm, which would hopefully make the description easier to follow.

---

> ### Author Response · Authors · 2025-11-27
> **Addressing the comment on the inference overhead due to "hundreds of reference problems"**
>
> We thank the reviewer for the thoughtful comment regarding the potential computational cost of using hundreds of reference problems for nearest-neighbor search. We agree that, in the absence of explicit analysis, this component can appear heavy.
>
> We respectfully direct the reviewer to our public comment *Runtime Analysis* in this thread, where we provide a detailed breakdown of the runtime for ProcessLID and all baselines, including how each component contributes to total inference cost.
>
> Here we summarize the key observations most relevant to this concern:
>
> 1. **Nearest-neighbor search is negligible relative to forward passes.**
>    In our experiments, we use 500 reference datapoints, which we found (Appendix A.4) to be a good trade-off between performance and efficiency. With this setup, the runtime overhead of nearest-neighbor search in ProcessLID is negligible. For all internal-representation–based methods (including ProcessLID), the dominant cost is the forward pass through the generator (and, for ProcessLID, the prover) to obtain embeddings. Thus, the primary runtime concern lies in the prover forward pass, not in the nearest-neighbor lookup itself.
>
> 2. **Prover overhead can be substantially reduced with small, aligned models.**
>    Although the additional forward pass through the prover model dominates ProcessLID’s overhead, we show in Sections 6.2 and 6.3 that this cost can be significantly reduced with minimal performance loss by using a smaller but aligned prover. In particular, the *ProcessLID Light* variant (using DS-QW-1.5B instead of DS-QW-14B as the prover model) presented in the public comment *Runtime Analysis* retains **98%** of the original ProcessLID’s performance while requiring only **41.5%** of its runtime (and is just **17%** slower than vanilla LID), and it still outperforms all other baselines.
>
> 3. **Overall runtime is comparable to other strong baselines.**
>    Our runtime study shows that ProcessLID has the same order of magnitude runtime as other baselines, with LN Entropy being substantially faster (≈7.7×) and CoT Entropy being substantially slower (≈4.2×). This indicates that, in practice, ProcessLID’s inference cost is modest rather than prohibitive.
>
> Taken together, these findings suggest that while hundreds of reference problems are indeed used, the associated nearest-neighbor search cost is neglegible compared to model forward passes, and the overall inference overhead of ProcessLID can be kept well within a practical range—especially when using smaller prover models that preserve most of the performance gains.

---

### Official Review · Reviewer_jim6 · 2025-11-09

**Soundness:** 3
**Presentation:** 3
**Contribution:** 2
**Rating:** 2
**Confidence:** 3

**Summary:**

The authors propose ProcessLID, a training-free step-level evaluation method for reasoning tasks. It combines the hidden-state LID dynamics of the Base model with a Prover model that estimates step correctness. The method is simple to use and doesn't require extra training for a variety of models. Experiments examine correlations between ProcessLID scores and step correctness across multiple reasoning benchmarks. In the experiments, ProcessLID shows consistently high performance over various settings.

**Strengths:**

- Proposes a simple, training-free approach for step-level evaluation of reasoning processes, which can be applied across different models without additional supervision.

- Demonstrates consistent correlation between ProcessLID scores and step correctness across multiple datasets and model scales, indicating good generalization of the signal.

**Weaknesses:**

[W1] Limited contribution beyond existing LID work. The idea of employing Local Intrinsic Dimension (LID) as a geometric indicator of confidence has already been explored and validated in prior work on QA-style and outcome-level tasks. This paper extends the concept to step-level reasoning, but this extension appears largely incremental rather than introducing a fundamentally new principle. If the aim is to adapt LID to the reasoning domain, the main challenge lies in handling long and unsegmented reasoning trajectories, where step boundaries are ambiguous or absent. In such cases, the proposed method’s reliance on explicit step segmentation is not well aligned with current reasoning paradigms. Demonstrating performance on long-form reasoning benchmarks such as AIME or other open-ended CoT settings would be required to substantiate the claimed reasoning-specific contribution.

[W2] Unfair baseline comparison due to missing Prover alignment. The paper compares ProcessLID against vanilla LID baselines that use only the Base model’s hidden states, while ProcessLID incorporates an additional Prover model (DS-QW-14B). This discrepancy introduces a confounding factor: the reported improvements may partly result from leveraging the richer representations of the DS-QW-14B Prover rather than from the proposed step-wise asymmetric formulation itself. To ensure fairness, the authors should evaluate vanilla LID under the same cross-model setup (i.e., using DS-QW-14B as the Prover) to isolate the effect of the formulation from that of the additional model.

[W3] Insufficient methodological justification for the asymmetric formulation. The paper argues that both the Base and Prover components are necessary, but provides limited empirical or theoretical justification for this design choice. The current ablations only evaluate partial removals (e.g., w/o Prover), without testing complementary configurations such as a Prover-only variant (using LID Embedding Delta alone) or a reversed-asymmetric setup where the roles of Base and Prover are exchanged. Such comparisons are essential to demonstrate that the proposed asymmetric combination is not arbitrary and that the claimed synergy between the two terms is empirically grounded. Without these results, it remains unclear whether the performance gains arise from the specific asymmetric formulation or simply from one dominant component.

[W4] Missing vanilla LID baseline in inference-time evaluation. In Section 5.2 (“Inference-time intervention for answer selection”), the paper compares ProcessLID with various reward and entropy-based baselines, yet omits the vanilla LID result. Since ProcessLID’s main claim is to improve over vanilla LID by introducing step-level and cross-model components, including this baseline under the same rollout selection setup is essential to quantify the actual benefit of the proposed formulation.

**Questions:**

-Did the vanilla LID baseline also use the DS-QW-14B Prover model? If not, does this difference raise fairness concerns in the comparison? (see W2)

- What are the results when using only the Prover term (LID Embedding Delta) or adopting the reversed-asymmetric configuration? Do these variants perform comparably to the proposed formulation? (see W3)

- What are the vanilla LID results under the same inference-time intervention setup? How much does ProcessLID improve over that baseline? (see W4)

---

> ### Author Response · Authors · 2025-11-27
> **Addressing [W1]**
>
> We thank the reviewer for raising this concern about the strength and reasoning-specific nature of ProcessLID’s contribution.
>
> First, we acknowledge that prior work has already established LID as a useful geometric confidence signal in QA-style and outcome-level settings. Our goal is not to introduce an entirely new geometric primitive, but to adapt LID to *process-level* reasoning by localizing correctness at each intermediate step and making it usable in step-wise interventions. To this end, ProcessLID departs from outcome-level LID usage in two key ways:
> - it operates on *local changes* in representation rather than absolute embeddings, via **Step LID Delta** and **LID Embedding Delta**, and
> - it uses an *asymmetric* cross-model formulation between a base generator and a prover model to sharpen step-local correctness signals.
>
> These changes are precisely motivated by multi-step reasoning: they are designed to detect when a trajectory starts drifting off a “good” manifold at a particular step, rather than only at the final answer.
>
> Regarding the reviewer’s point about long and potentially unsegmented trajectories: we agree that handling such trajectories is a central challenge for reasoning-oriented methods. While our current experimental setup does not explicitly target AIME or other fully open-ended CoT benchmarks, the ProcessBench datasets already include highly non-trivial, multi-step math reasoning problems. In particular, **OmniMath** and **OlympiadBench** contain a substantial portion of Olympiad-level questions, whose difficulty is at least comparable to (and often above) AIME-style problems. ProcessLID’s consistent gains on these two sub-datasets—as well as on the grade-school level subsets—suggest that the method is robust across a wide spectrum of reasoning difficulty, from relatively short derivations to very challenging, multi-step proofs.
>
> We agree, however, that our original treatment of step segmentation in the inference-time experiments with fixed length of 100 tokens was not ideally aligned with current reasoning paradigms. This issue was also flagged by other reviewers. In response, we have *updated* the inference-time setup for ProcessLID to split rollouts at the delimiter `\n\n`. This segmentation scheme yields steps that are more semantically self-contained, which is better aligned with modern reasoning paradigms.
>
> The updated inference-time results are reported in Table 3 (Section 5.2). We present the key observations here:
> - On *Oracle@N (Pruned-5)* and *Weighted Majority Vote*, ProcessLID now **outperforms** the reward model for both generator models by an average of **2.00 AUROC** and **0.33 AUROC**, respectively.
> - On *Best of N*, ProcessLID trails the reward model by only **1.82 AUROC** on average, a substantial improvement over the original fixed-length setup (where the gap was 5.66 AUROC).
>
> We emphasize that the reward model baseline is a high-capacity 8B sequence scorer trained explicitly to rate full sequences using substantial human preference data and compute. By contrast, ProcessLID is fully training-free and relies only on internal representations plus a calibration set. In this context, (i) outperforming the reward model on two inference-time selection metrics and (ii) remaining highly competitive on *Best of N* under a semantically aligned segmentation scheme provides concrete evidence that ProcessLID is effective in realistic reasoning without explicit step segmentations.
>
> In summary, the current results showed that (a) ProcessLID’s design handles challenging, multi-step math reasoning (including Olympiad-level problems) and (b) it remain competitive against a strong, trained reward model in inference-time experiment without explicit step segmentations. We see these findings as a meaningful and non-trivial extension of LID from outcome-level QA to process-level reasoning, and we also acknowledge that long-form open-ended benchmarks is a natural next step for future work.

---

> ### Author Response · Authors · 2025-11-27
> **Addressing [W2] and [W3] (and their corresponding questions)**
>
> We thank the reviewer for these thoughtful comments regarding (i) the fairness of our comparison to vanilla LID when ProcessLID uses an additional Prover model, and (ii) the need for stronger justification of our asymmetric formulation.
>
> We acknowledge that in the original ablation study, the vanilla LID baseline did **not** use the DS-QW-14B Prover model, and that this made it harder to fully disentangle the effect of our formulation from the effect of introducing a stronger model. In response, we have expanded our ablation study to include exactly the configurations suggested in [W2] and [W3]:
>
> 1. **Vanilla w/ Prover** – a cross-model variant of vanilla LID that uses the same base–prover pairing as ProcessLID. This variant computes vanilla LID for both the base and Prover models, thereby aligning representational capacity and isolating the contribution of our step-wise asymmetric formulation.
>
> 2. **Only Prover** – a Prover-only variant that evaluates only the Prover term in ProcessLID (using LID Embedding Delta), effectively testing whether the Prover alone dominates performance.
>
> 3. **Reversed** – a variant that swaps the roles of the base and Prover terms in the asymmetric formulation, directly probing whether our particular choice of which model contributes which signal is crucial, or whether any asymmetric combination would suffice.
>
> We respectfully direct the reviewer to our public comment *Addressing the Lack of Ablation Study* in this thread, where we present the full quantitative results and detailed analysis for these variants.
>
> In summary, the expanded ablation study shows that:
> - Simply giving vanilla LID access to the same strong Prover (*Vanilla w/ Prover*) does **not** close the gap to ProcessLID,
> - Prover-only, reversed-asymmetric, and symmetric variants are consistently weaker than our proposed asymmetric formulation, and
> - ProcessLID’s specific asymmetric combination of base and Prover signals yields the strongest and most stable performance.
>
> We also emphasize that Section 6.4 provides a theoretical justification for ProcessLID’s asymmetric design. There, we introduce the notion of alignment, defined as the mutual information between (i) the residual information contributed by the Prover term that is not captured by the base model term, and (ii) the ground-truth label of each reasoning step (a rigorous formalization and empirical validation of this quantity are given in Appendix A.9). Intuitively, a higher alignment means the Prover is supplying more label-relevant (i.e., correctness-relevant) information beyond what the Base model already provides.
>
> From this perspective, we hypothesized that the asymmetric formulation should outperform symmetric ones (where both Base model and Prover use either Step LID Delta or LID Embedding Delta) because it better positions the Prover to contribute additional, correctness-relevant signal on top of the Base model—i.e., it should yield higher alignment.
>
> Table 5 reports both the alignment and step-level AUROC on ProcessBench for ProcessLID and its two symmetric variants. As shown in the table, alignment and AUROC move together: ProcessLID achieves the highest alignment and, correspondingly, the highest AUROC, while the symmetric variants exhibit both lower alignment and lower AUROC. This strong correlation between alignment and performance provides a principled, theoretic explanation for why the asymmetric formulation is preferable.
>
> Taken together, these findings offer both empirical and theoretical support that directly address the fairness concern in [W2] and the design question in [W3]. They show that our gains are not merely a by-product of the Prover’s raw capacity or a single dominant component, but instead arise from the synergy between the asymmetric design, layer-wise aggregation, and the way ProcessLID explicitly isolates and leverages the Prover’s distinct representation geometry.

---

> ### Author Response · Authors · 2025-11-27
> **Addressing [W4] (and its correspoding question)**
>
> We thank the reviewer for pointing out the missing vanilla LID baseline in our inference-time evaluation [W4]. We agree that including vanilla LID under the same rollout selection setup is essential for quantifying the actual benefit of ProcessLID.
>
> In the updated paper, we now include vanilla LID as a baseline in the inference-time experiments. For both ProcessLID and vanilla LID, we evaluate compute the step-level signals on semantically self-contained steps obtained by segmenting rollouts using the delimiter `\n\n` (rather than fixed-length token chunks), and we apply the same rollout selection procedures across all methods. The full results are reported in Table 3 in Section 5.2.
>
> We highlight that, under this shared setup, ProcessLID outperforms vanilla LID on all three evaluation metrics—Oracle@N (Pruned-5), Weighted Majority Vote, and Best of N—with average margins of 3.67, 0.66, and 4.96 AUROC, respectively. These improvements directly quantify the benefit of the proposed step-level, cross-model formulation over the vanilla LID baseline in the inference-time intervention setting.

---

### Official Review · Reviewer_XCVa · 2025-11-14

**Soundness:** 3
**Presentation:** 2
**Contribution:** 2
**Rating:** 4
**Confidence:** 3

**Summary:**

The paper proposes ProcessLID, a training-free, representation-based method for estimating step-level correctness in math chain-of-thought (CoT) reasoning using local intrinsic dimension (LID). It computes a Step LID Delta signal from the base model and a complementary LID Embedding Delta signal from a separate prover model, then aggregates layer-wise scores via calibrated p-values and Fisher’s method. Experiments on four ProcessBench math datasets and an outcome-level MATH-CoT dataset show that ProcessLID improves step- and outcome-level AUROC over entropy-based uncertainty and prior representation-based baselines.

**Strengths:**

1. The paper clearly motivates step-level correctness estimation in math CoTs as a practically important problem and uses local intrinsic dimension in a conceptually grounded, easy-to-follow way.
2. ProcessLID is a simple, training-free approach that combines base-model and prover LID signals with a calibrated multi-layer aggregation scheme that is straightforward to implement once embeddings are available.
3. Experiments across four math benchmarks, six generator models, and an outcome-level MATH-CoT dataset show consistent AUROC gains over strong uncertainty- and representation-based baselines, supported by ablations, prover-choice studies, and neighborhood-size analyses.

**Weaknesses:**

1. Although the method is “training-free,” it relies on step-level labeled calibration sets for layer selection and p-value normalization, and the sample efficiency and cross-dataset/model transfer of this calibration are not systematically analyzed.
2. The inference-time Best-of-N experiments are limited to 150 MATH-500 questions and two generators, and the gains over a strong reward model are modest, so the end-to-end impact on problem-solving performance is still somewhat preliminary.
3. The evaluation does not fairly isolate the method’s contribution: ProcessLID uses a strong prover while vanilla LID does not, and missing variants (Prover-only, reversed roles, cross-model baselines) make the asymmetric design insufficiently justified.

**Questions:**

How well do the chosen layers, p-value mappings, and mixing weight α transfer across datasets and models, and how much step-level labeled calibration data is needed before performance degrades noticeably?

---

> ### Author Response · Authors · 2025-11-27
> **Addressing the sample efficiency and cross-dataset/model transfer of hyperparameters and p-value normalization**
>
> We thank the reviewer for highlighting the importance of sample efficiency and cross-dataset/model transfer for the calibration procedure (layer selection, p-value mapping, and the mixing weight $\alpha$). These aspects were indeed underexplored in the original submission, and we agree that they are essential for demonstrating the robustness and practicality of ProcessLID.
>
> To address this, we added new experiments in Appendix A.3.1 that systematically study the effect of calibration set size on ProcessLID’s performance (Figure 4). For each ProcessBench sub-dataset (noting that each sub-dataset corresponds to a distinct question source with different difficulty and CoT style), we vary the calibration dataset size (25, 50, then in increments of 50 up to the full dataset) and, for each size, recompute the layer-wise $p$-values and hyperparameters. We then measure the step-level AUROC for each base model.
>
> Across all four sub-datasets, we observe that performance quickly stabilizes once the calibration set reaches roughly 150–200 examples: beyond this range, enlarging the calibration set yields negligible gains. Noticeable degradation only appears when the calibration set is smaller than 100–150 examples. This indicates that our layer selection, $p$-value normalization, and $\alpha$ selection are highly sample-efficient and do not require large labeled calibration sets to achieve near-optimal performance.
>
> We further analyze the transferability of calibrated hyperparameters in Appendix A.3.2, with the main results summarized in Tables 6 and 7.
>
> - **Transfer across datasets.** For each base model and each “source” sub-dataset in ProcessBench, we first calibrate the hyperparameters on that dataset. We then transfer these hyperparameters to all *other* sub-datasets while keeping the base model fixed and evaluate step-level AUROC. The table below (averaged over all base models) reports how well calibration on one dataset generalizes to others. Row names indicate the dataset used for calibration; column names indicate the target dataset.
>
> | | ProcessBench - GSM8K | ProcessBench - MATH | ProcessBench - OlympiadBench | ProcessBench - OmniMath |
> |:-----------------------------|-----------------------:|----------------------:|-------------------------------:|--------------------------:|
> | ProcessBench - GSM8K | 90.01 | 86.62 | 83.81 | 85.72 |
> | ProcessBench - MATH | 80.39 | 88.96 | 83.51 | 83.48 |
> | ProcessBench - OlympiadBench | 78.22 | 80.96 | 86.46 | 82.17 |
> | ProcessBench - OmniMath | 78.27 | 79.77 | 81.83 | 87.39 |
>
> - **Transfer across models.** Symmetrically, for each sub-dataset and each “source” base model, we calibrate the hyperparameters on that model, then transfer them to all *other* base models on the same dataset and re-evaluate step-level AUROC. The table below (averaged over all datasets) reports how well calibration on one model generalizes to others. Row names now indicate the model used for calibration; column names indicate the target model.
>
> | | DS-LLAMA-8B | DS-QW-1.5B | DS-QW-14B | DS-QW-32B | DS-QW-7B | LLAMA-3.1-8B |
> |:---------------|--------------:|-------------:|------------:|------------:|-----------:|---------------:|
> | DS-LLAMA-8B | 86.49 | 83.55 | 86.11 | 85.39 | 86.5 | 84.77 |
> | DS-QW-1.5B | 82.07 | 87.78 | 83.09 | 82.69 | 85.32 | 84 |
> | DS-QW-14B | 81.87 | 79.78 | 90.31 | 85.73 | 83.01 | 84.42 |
> | DS-QW-32B | 83.56 | 80.6 | 87.87 | 88.34 | 85.47 | 85.45 |
> | DS-QW-7B | 81.19 | 83.22 | 83.61 | 83.02 | 88.56 | 83.27 |
> | LLAMA-3.1-8B | 81.13 | 81.33 | 83.67 | 82.47 | 83.09 | 87.74 |
>
> From the aggregated summaries in Tables 6 and 7, we find that:
> - Transferring hyperparameters **across datasets** leads to an average performance drop of only $6.14$ AUROC.
> - Transferring hyperparameters **across models** leads to an average performance drop of only $4.62$ AUROC.
>
> In both cases, the degradation is modest relative to the absolute performance levels and far smaller than the margins by which ProcessLID improves over strong baselines.
>
> Taken together, the calibration-size study (Appendix A.3.1) and the cross-dataset/model transfer experiments (Appendix A.3.2) provide concrete evidence that:
> 1. The calibration procedure is **sample-efficient**, requiring only ~150–200 labeled steps to reach near-optimal performance, and
> 2. The resulting layers, $p$-value mappings, and mixing weight $\alpha$ are **robust and transferable** across both datasets and models, with only small performance degradation.

---

> ### Author Response · Authors · 2025-11-27
> **Addressing the lack of end-to-end impact on problem solving performance**
>
> We thank the reviewer for this thoughtful comment on the inference-time evaluation and for pointing out that the original gains over a strong reward model were modest.
>
> In revising the paper, and in response to concerns from multiple reviewers, we refined the inference-time setup for ProcessLID to better align with the method’s design. Specifically:
>
> 1. **Semantically aligned step segmentation.** Instead of segmenting rollouts into fixed-length steps of 100 tokens, we now segment steps using the delimiter `\n\n`. This produces steps that are more semantically self-contained (e.g., roughly corresponding to natural reasoning units), which is more compatible with our Step LID Delta and LID Embedding Delta formulations.
>
> 2. **Inclusion of Vanilla LID in rollout evaluation.** We now include Vanilla LID as an additional baseline in the inference-time experiments, evaluated under the same step segmentation and scoring protocol as ProcessLID.
>
> The updated inference-time results are reported in Table 3 (Section 5.2). We present the critical observations here:
> - On *Oracle@N (Pruned-5)* and *Weighted Majority Vote*, ProcessLID now **outperforms** the reward model for both generator models by an average of 2.00 and 0.33 AUROC, respectively.
> - On *Best of N*, ProcessLID trails the reward model by only 1.82 points on average (compared to a gap of 5.66 points in the original fixed-length setup).
>
> We emphasize that the reward model is a high-capacity 8B sequence scorer trained with substantial human preference data and compute, whereas ProcessLID is entirely training-free and uses only internal representations. In this context, surpassing the reward model on two metrics and remaining highly competitive on *Best of N*—without any task-specific training—provides stronger evidence that ProcessLID has meaningful end-to-end impact on problem-solving accuracy.
>
> For clarity, we also compare ProcessLID’s performance under the original and revised step segmentation schemes (displayed below). Moving from fixed 100-token steps to `\n\n`-delimited steps yields consistent improvements across all three metrics: +2.66 (Oracle@N Pruned-5), +1.22 (Weighted Majority Vote), and +3.84 (Best of N). This behavior is aligned with our design intuition: Step LID Delta and LID Embedding Delta are intended to isolate the marginal correctness contribution of each reasoning step, which is more faithfully captured when steps are semantically coherent rather than arbitrary token slices.
>
> |                           | LLAMA-3.1-8B         | DS-LLAMA-8B          | Average        |
> |---------------------------|----------------------|----------------------|----------------|
> | **Oracle @ N Pruned 5**   |                      |                      |                |
> | ProcessLID (Old)          | $$74.89_{0.36}$$     | $$81.78_{0.36}$$     | $$78.34$$      |
> | ProcessLID (New)          | $$78.00_{3.20}$$     | $$84.00_{1.85}$$     | $$81.00$$      |
> | **Weighted Majority Vote**|                      |                      |                |
> | ProcessLID (Old)          | $$62.22_{1.28}$$     | $$80.00_{0.62}$$     | $$71.11$$      |
> | ProcessLID (New)          | $$65.33_{3.20}$$     | $$79.33_{1.85}$$     | $$72.33$$      |
> | **Best of N**             |                      |                      |                |
> | ProcessLID (Old)          | $$54.67_{1.23}$$     | $$69.56_{0.36}$$     | $$62.12$$      |
> | ProcessLID (New)          | $$57.33_{2.13}$$     | $$74.59_{2.97}$$     | $$65.96$$      |
>
> While the inference-time experiments are still conducted on 150 MATH-500 questions and two generators (to keep compute manageable), the improvements under the semantically grounded setup, the stronger comparison against a state-of-the-art reward model, and the inclusion of Vanilla LID as a rollout baseline together provide a more convincing and concrete demonstration of ProcessLID’s end-to-end impact in inference-time settings.

---

> ### Author Response · Authors · 2025-11-27
> **Addressing the unfair comparison between ProcessLID and Vanilla LID and insufficient justification of the asymmetric formulation**
>
> We thank the reviewer for this helpful comment. We agree that the original ablations did not fully disentangle the contribution of the prover model from the contribution of our asymmetric formulation.
>
> We have since expanded our ablation study to include exactly the variants the reviewer requested:
> 1. **Vanilla w/ Prover** – a cross-model variant of vanilla LID that uses the same base–prover pairing as ProcessLID. This variant computes vanilla LID for both the base and Prover models, thereby aligning representational capacity and isolating the contribution of our step-wise asymmetric formulation.
>
> 2. **Only Prover** – a Prover-only variant that evaluates only the Prover term in ProcessLID (using LID Embedding Delta), effectively testing whether the Prover alone dominates performance.
>
> 3. **Reversed** – a variant that swaps the roles of the base and Prover terms in the asymmetric formulation, directly probing whether our particular choice of which model contributes which signal is crucial, or whether any asymmetric combination would suffice.
>
> We respectfully direct the reviewer to our public comment *Addressing the Lack of Ablation Study* in this thread, where we present the full quantitative results and detailed analysis for these variants.
>
> In brief, these new ablations show that (1) simply adding a strong prover to vanilla LID does **not** close the performance gap to ProcessLID, and (2) Prover-only, reversed-asymmetric, and symmetric variants remain noticeably weaker than ProcessLID's specific asymmetric formulation. Together, this indicates that ProcessLID’s gains are not driven solely by the prover’s raw capacity, and that our specific asymmetric combination of base and prover signals is empirically important for its performance.

---

### Author Response · Authors · 2025-11-27
**Addressing the Lack of Ablation Study**

Multiple reviewers raised concerns that the original ablation study did not:

1. Clearly demonstrate that ProcessLID’s gains are **not** purely due to the richer representations of a stronger Prover model; and
2. Provide sufficient empirical support for the **asymmetric formulation** that combines Step LID Delta for the Base model with LID Embedding Delta for the Prover.

To address these points, we have updated our ablation study to include the following variants:

1. *w/o Aggregation*: replaces Fisher’s aggregation over $\mathcal{L}^*$ with a single best-performing layer.
2. *Vanilla w/ Prover*: computes vanilla LID for both Base and Prover terms, $R(E_{\pi,j}) = m_{\pi}(E_{\pi,j}) + \alpha\, m_{\mu}(E_{\mu,j})$, thereby giving vanilla LID access to the **same Base–Prover pair** as ProcessLID, but without deltas or layer-wise aggregation.
3. *Only Prover*: uses only the Prover term,  $R(E_{\pi,j}) = m_{\mu}(\Delta E_{\mu,j})$, isolating the contribution of the Prover’s LID Embedding Delta alone.
4. *Reversed*: swaps the roles of Base and Prover, $R(E_{\pi,j}) = \Delta m_{\mu}(E_{\pi,j}) + \alpha\, m_{\pi}(\Delta E_{\pi,j})$, to test whether the particular asymmetric choice in ProcessLID is crucial.

The detailed numbers are reported in Table 6 (Section 6.1) of the paper. Here we summarize how these new variants address the reviewers’ concerns.

## On fairness and the role of the Prover.
The *Vanilla w/ Prover* variant exhibits an average drop of 3.14 AUROC relative to ProcessLID. Importantly, *Vanilla w/ Prover* uses **exactly the same Base and Prover models** as ProcessLID; the only difference is that it applies vanilla LID to both, without our delta-based design or layer aggregation. This provides a fair, cross-model baseline for vanilla LID and shows that merely adding a strong Prover does **not** recover ProcessLID’s performance. In other words, the Prover’s raw representational capacity alone cannot explain the gains.

Similarly, the *Only Prover* variant, which uses only the Prover’s LID Embedding Delta, shows an average drop of 2.66 AUROC. This indicates that even when we meaningfully capture the Prover’s representation geometry via $m_{\mu}(\Delta E_{\mu,j})$, that signal by itself is insufficient to match ProcessLID. The best performance arises when this Prover term is combined asymmetrically with the Base model’s Step LID Delta and layer-wise aggregation. Together, *Vanilla w/ Prover* and *Only Prover* directly address the concern that ProcessLID might simply be “winning by using a stronger model”: the data show that the Prover is helpful, but **ProcessLID’s formulation and aggregation are essential** to fully realize that benefit.

## On the necessity of the asymmetric formulation.
The *Reversed* variant, which swaps the roles of Base and Prover, suffers an average drop of 2.13 AUROC compared to ProcessLID. This shows that the specific asymmetric choice in ProcessLID—Base contributing Step LID Delta and Prover contributing LID Embedding Delta—is not arbitrary: exchanging these roles yields consistently weaker performance.

We further analyze symmetry vs. asymmetry in Table 5 (Section 6.4), which compares ProcessLID against two fully symmetric formulations:

| **Formulation / Model**      | **LLAMA-3.1-8B** | **DS-QW-1.5B** | **DS-QW-7B** | **DS-LLAMA-8B** | **DS-QW-14B** | **DS-QW-32B** | **_Average_** |
|------------------------------|-----------------:|---------------:|-------------:|----------------:|--------------:|--------------:|--------------:|
| Both Step LID Delta          | 86.94            | 82.14          | 82.77        | 79.64           | 80.11         | 83.95         | 82.59 ($-5.53$) |
| Both LID Embedding Delta     | 85.97            | 86.03          | 86.70        | 84.37           | 85.81         | 86.03         | 85.82 ($-2.30$) |
| **ProcessLID (Original)**    | **89.41**        | **87.03**      | **87.73**    | **85.36**       | **89.74**     | **89.46**     | **88.12**     |

Relative to ProcessLID, *Both LID Embedding Delta* incurs an average drop of 2.30 AUROC, and *Both Step LID Delta* incurs a larger drop of 5.53 AUROC. These results show that **symmetric formulations are consistently less effective** than ProcessLID’s asymmetric design. When combined with the *Reversed* and *Only Prover* variants, the picture is clear: the specific asymmetric combination of Base Step LID Delta and Prover LID Embedding Delta is empirically superior to (i) using only one component of ProcessLID, (ii) simply adding a strong Prover to vanilla LID, or (iii) any symmetric variants.

---

### Author Response · Authors · 2025-11-27
**Runtime Analysis [1/2]**

Several reviewers noted that the runtime and practical overhead of ProcessLID were not sufficiently discussed. In particular, they raised two main concerns:
1. Computing LID via nearest-neighbor search might incur substantial runtime cost.
2. The additional forward pass through the Prover model required for the LID Embedding Delta might introduce significant overhead.

To address these concerns, we conducted a runtime analysis of ProcessLID and several baselines. We define **runtime per reasoning step** as the wall-clock time (in seconds) between receiving a single reasoning step as input and producing its correctness score. All methods are evaluated using LLAMA-3.1-8B as the generator model on ProcessBench. For ProcessLID, we use DS-QW-14B as the Prover model. For both ProcessLID and Vanilla LID, we use 500 reference data points for the nearest-neighbor search, which is the average reference size that achieves near-optimal performance with good efficiency (as analyzed in Appendix A.4).

The table below reports the runtime per reasoning step (second column; subscript indicate the 95% confidence interval)  and the average step-level AUROC on ProcessBench (third column). The remaining columns indicate which computational components contribute to each method’s runtime. The “Method-Specific Operation” column lists the core computation applied once all required inputs have been obtained.

| Method                     | Runtime (s) per reasoning step | ProcessBench avg. step-level AUROC | Forward pass in generator model                                            | Forward pass in prover model                 | External judge LLM                                                           | Nearest-Neighbor search                      | Method-Specific Operation |
|----------------------------|--------------------------------|------------------------------------|----------------------------------------------------------------------------|----------------------------------------------|-------------------------------------------------------------------------------|----------------------------------------------|---------------------------|
| LN Entropy                 | $0.508_{0.212}$               | 71.78                              | Yes, a single forward pass to extract the logit distribution               |                                              |                                                                               |                                              | Compute entropy           |
| Chain of Embeddings        | $1.83_{0.0933}$               | 71.44                              | Yes, a forward pass for every token in the reasoning step to extract embeddings |                                              |                                                                               |                                              | Compute CoE score         |
| CoT Entropy                | $16.2_{5.83}$                 | 82.92                              |                                                                            |                                              | Yes, requires a judge LLM to generate a full CoT response for the reasoning step |                                              | Compute entropy           |
| ProcessLID                 | $3.90_{1.63}$                 | 89.25                              | Yes, a single forward pass to extract embeddings                           | Yes, a single forward pass to extract embeddings |                                                                               | Both for generator model and Prover model    | Compute LID               |
| ProcessLID Light           | $1.62_{0.668}$                | 87.45                              | Yes, a single forward pass to extract embeddings                           | Yes, a single forward pass to extract embeddings |                                                                               | Both for generator model and Prover model    | Compute LID               |
| Vanilla LID                | $1.38_{0.580}$                | 79.15                              | Yes, a single forward pass to extract embeddings                           |                                              |                                                                               | Generator model                               | Compute LID               |

We direct the viewer to our subsequent public comment "Runtime Analysis [2/2]" for the detailed analysis of these results.

---

### Author Response · Authors · 2025-11-27
**Runtime Analysis [2/2]**

First, we note that the **runtime per reasoning step for ProcessLID (3.90s)** is of the same order of magnitude as other strong methods (with the exception of CoT Entropy, which is substantially slower), indicating that its overall inference cost is moderate rather than prohibitive.

Second, for all **internal-representation-based methods** (LN Entropy, Chain of Embeddings, ProcessLID, and Vanilla LID), the runtime is dominated by one or more **forward passes through the generator (and, for ProcessLID, the Prover) model(s)**. Once embeddings or logits are available, the additional computations—such as nearest-neighbor search or entropy/LID calculation—contribute neglegibly little to the total runtime. In particular, the nearest-neighbor search in ProcessLID is implemented with FAISS, a highly optimized library for vector similarity search: the index is built once, and subsequent nearest-neighbor queries are extremely efficient. Empirically, the cost of the FAISS-based nearest-neighbor search is negligible relative to the model forward passes.

Third, we directly address the overhead of the Prover model. As expected, the forward pass through the Prover dominates the runtime of ProcessLID and scales roughly linearly with the Prover’s size. However, as analyzed in Section 6.2, our results show that **even small Prover models can yield substantial performance gains over Vanilla LID**. This means we can significantly reduce the Prover overhead by choosing a smaller, well-aligned Prover with minmal performance trade-offs. The *ProcessLID Light* variant in the table concretely illustrates this point: using DS-QW-1.5B as the Prover, it achieves **87.45** step-level AUROC with a runtime of **1.62s** per reasoning step. This retains **98%** of the original ProcessLID’s performance while using only **41.5%** of its runtime (and is only about **17%** slower than Vanilla LID). Moreover, *ProcessLID Light* still outperforms all other baselines except the full ProcessLID, demonstrating that Prover overhead can be substantially reduced while preserving nearly optimal accuracy.

Finally, from a performance–cost perspective, CoT Entropy attains the strongest performance among the non-ProcessLID baselines (82.92 AUROC) but requires **over 4×** the runtime of ProcessLID (and about **10×** that of ProcessLID Light), due to the full CoT generation from a judge LLM. In contrast, while ProcessLID and ProcessLID Light are somewhat slower than other internal-signal baselines, the **accuracy gains are substantial**. This shows that ProcessLID can effectively convert a reasonable increase in inference-time computation into a large improvement in reasoning correctness.

---

### Comment · Area_Chair_2BNT · 2025-11-28

Dear Reviewers,

Thank you for your time and efforts in serving as a reviewer.

The authors have submitted their rebuttal, and this AC kindly asks you to review their response and assess whether your comments have been adequately addressed. If any points require further clarification or discussion, please feel free to raise those questions by adding comments and initiating discussion as needed.

ICLR encourages reviewers to actively engage in the discussion phase, so your prompt actions are especially valuable. Thank you very much for your continued efforts and valuable contributions.

Best regards,

Your AC

---

### Author Response · Authors · 2025-12-03
**Note to AC [1/3]**

Given the recent change in the rebuttal procedure, we use this comment as an aggregation post to summarize the major concerns raised by reviewers and how we have addressed them in the revised paper and public comments.

## 1. Sample efficiency and transferability of calibrated hyperparameters and p-value normalization

Reviewer **XCVa** raised the concern that, although ProcessLID is training-free, the *sample efficiency* and *cross-dataset/model transfer* of the calibration procedure (for layer selection, p-value normalization, and the mixing weight $\alpha$ were not systematically analyzed.

To address this, we added new experiments in **Appendix A.3.1** that systematically study:
- how the size of the calibration dataset affects ProcessLID’s performance, and
- how calibrated hyperparameters transfer across models and datasets.

For a detailed discussion, we respectfully direct the AC to this [comment](https://openreview.net/forum?id=5O5AlNVAbs&noteId=wwHAqvPNwr). In summary:

- **Sample efficiency.** When varying the calibration set size, we find that ProcessLID requires only **150–200** labeled calibration points to reach near-optimal step-level AUROC. Performance is stable once the calibration size exceeds this range, and noticeable degradation appears only when the calibration set drops below **100–150** examples. This suggests that our p-value normalization and hyperparameters are **highly sample efficient**.

- **Transferability.** Transferring hyperparameters *across datasets* leads to an average drop of only **6.14 AUROC**, and transferring them *across models* leads to an average drop of **4.62 AUROC**. Given the absolute performance levels, these degradations are modest. This indicates that the calibrated hyperparameters and p-value normalization are **robust and transferable** across both datasets and base models.

---

## 2. End-to-end impact on problem-solving performance

Multiple reviewers (**XCVa, jim6, NGTy, NmXq**) raised concerns about the *end-to-end* impact of ProcessLID on problem-solving performance, pointing to:

1. modest gains over a strong supervised reward model and the missing **Vanilla LID** baseline in inference-time experiments, and
2. the use of fixed-length (100-token) step segmentation, which does not align with modern reasoning paradigms where step boundaries are semantic and variable.

To address these concerns, we updated our inference-time setup in Section 5.2 as follows:

- We now segment rollouts using the delimiter `\n\n`, producing **semantically self-contained steps** that better reflect natural reasoning units.
- We include **Vanilla LID** as a baseline under the *same* rollout selection setup as ProcessLID.

For full details, we direct the AC to this [comment](https://openreview.net/forum?id=5O5AlNVAbs&noteId=o4lX93fx9E). In summary:

- **Improvement over Vanilla LID.** ProcessLID outperforms Vanilla LID on all three inference-time metrics, with average margins of:
  - **+3.67 AUROC** on *Oracle@N (Pruned-5)*,
  - **+0.66 AUROC** on *Weighted Majority Vote*, and
  - **+4.96 AUROC** on *Best of N*.

  These margins directly quantify the added value of ProcessLID over its vanilla counterpart in the inference time experiment setting.

- **Competitiveness vs. supervised reward model.** Under the updated (semantic) step segmentation:
  - On *Oracle@N (Pruned-5)* and *Weighted Majority Vote*, ProcessLID **outperforms** the supervised reward model for both generator models by an average of **2.00** and **0.33 AUROC**, respectively.
  - On *Best of N*, ProcessLID trails the reward model by only **1.82 AUROC** on average (improved from a **5.66 AUROC** gap in the original fixed-length step segmentation setup).

Given that the reward model is a high-capacity 8B sequence scorer trained with substantial data and compute, **surpassing it on two metrics and remaining highly competitive on Best of N—without any task-specific training—provides strong evidence that ProcessLID meaningfully improves end-to-end problem-solving performance** in realistic inference-time scenarios.

---

---

### Author Response · Authors · 2025-12-03
**Note to AC [2/3]**

## 3. Lack of comprehensive ablation study

Multiple reviewers (**XCVa, jim6**) raised concerns about the completeness of our ablations. In particular:

- ProcessLID was not compared to a **Vanilla LID** variant that also uses a prover model, making it unclear whether gains stemmed primarily from the prover’s stronger representational power.
- The **asymmetric formulation** in ProcessLID was not fully justified, due to missing variants that swap or isolate the roles of the base and prover components.

To address this, we expanded our ablation study to include exactly the requested variants:

1. **Vanilla w/ Prover:**
   $$
   R(E_{\pi,j}) = m_{\pi}(E_{\pi,j}) + \alpha m_{\mu}(E_{\mu,j})
   $$
   This variant computes vanilla LID for both the base and Prover models, thereby giving Vanilla LID access to the **same Base–Prover pair** as ProcessLID and isolating the contribution of our step-wise asymmetric formulation. (This variant was originally reported as **w/o Deltas**; we now rename it **Vanilla w/ Prover** for clarity.)

2. **Only Prover:**
   $$
   R(E_{\pi,j}) = m_{\mu}(\Delta E_{\mu,j})
   $$
   A Prover-only variant that uses only the prover term (LID Embedding Delta), testing whether the prover alone can explain the gains.

3. **Reversed:**
   $$
   R(E_{\pi,j}) = \Delta m_{\mu}(E_{\mu,j}) + \alpha m_{\pi}(\Delta E_{\pi,j})
   $$
   A variant that swaps the roles of the base and prover in the asymmetric formulation, probing whether **any** asymmetric combination works or whether our particular assignment is crucial.

Additionally, we compare ProcessLID to **symmetric variants** (both terms using Step LID Delta or both using LID Embedding Delta) in Section 6.3, and we provide a **theoretical justification** for the asymmetric design in Section 6.4.

For full details, we direct the AC to our public comment *Addressing the Lack of Ablation Study*. In brief, the expanded ablations show that:

- Simply adding a strong prover to Vanilla LID (**Vanilla w/ Prover**) does **not** close the performance gap to ProcessLID.
- Prover-only, reversed-asymmetric, and fully symmetric variants are **consistently weaker** than ProcessLID’s original asymmetric formulation.
- ProcessLID’s gains are **not dominated by any single component**; they arise from the synergy between:
  - the asymmetric design (Step LID Delta for the base, LID Embedding Delta for the prover),
  - layer-wise aggregation, and
  - capturing the prover’s distinct representation geometry.

Taken together, this addresses the fairness and justification concerns and supports the conclusion that **ProcessLID’s improvements are not driven solely by the prover’s raw capacity, but by its specific, empirically grounded asymmetric combination of base and prover signals.**

---

### Author Response · Authors · 2025-12-03
**Note to AC [3/3]**

## 4. Inference-time overhead of ProcessLID

Multiple reviewers (**kquz, NGTy, NmXq**) raised concerns about the potential inference-time overhead of ProcessLID, due to:

1. nearest-neighbor search over reference datapoints when computing LID, and
2. the additional forward pass through the prover model.

To address this, we conducted a detailed runtime analysis of ProcessLID and all baselines, decomposing the total inference cost into its constituent operations. For complete results, we refer the AC to our public comment *Runtime Analysis*. Here we summarize the key findings:

1. **Nearest-neighbor search is negligible compared to model forward passes.**
   In our experiments, we use **500 reference datapoints**, which we identified (Appendix A.4) as a good trade-off between performance and efficiency. Under this configuration, the nearest-neighbor search contributes a neglegible fraction of total runtime. For all internal-representation-based methods (including ProcessLID), the **dominant cost** is the forward pass through the generator (and, for ProcessLID, the prover) to obtain embeddings. Thus, the primary overhead lies in the prover forward pass, not the nearest-neighbor lookup.

2. **Prover overhead can be substantially reduced with small, aligned models.**
   While the additional forward pass through the prover model does introduce extra cost, we show in Sections 6.2 and 6.3 that this overhead can be substantially reduced with minimal performance loss by using a **smaller but aligned** prover. In particular, the **ProcessLID Light** variant in the public comment *Runtime Analysis* (using DS-QW-1.5B instead of DS-QW-14B as the prover):
   - retains **98%** of the original ProcessLID’s performance,
   - uses only **41.5%** of its runtime, and
   - is just **17% slower** than Vanilla LID,
   while still outperforming all non-ProcessLID baselines.

3. **Overall runtime is comparable to other strong baselines.**
   Our runtime study shows that ProcessLID has the **same order of magnitude** runtime as other strong baselines:
   - LN Entropy is ≈7.7× faster, and
   - CoT Entropy is ≈4.2× slower.

   This indicates that ProcessLID’s inference-time cost is **modest rather than prohibitive**.

Overall, these results show that the main additional cost comes from a single prover forward pass, which can be substantially reduced by using smaller aligned provers, and that the nearest-neighbor search overhead is negligible in practice.

---

## 5. Guidance and robustness for a priori prover model selection

Reviewers **NGTy** and **NmXq** raised several related concerns regarding the prover model:

1. Whether ProcessLID’s performance is *highly sensitive* to the choice of prover model.
2. Whether using a much weaker prover, or having a large capability gap between base and prover, can negatively affect alignment and performance.
3. The lack of *practical guidance* on how to select a prover *a priori* without extensive experimentation.

To address the first two concerns, we point to the prover ablation in Section 6.2 and a detailed analysis in this [comment](https://openreview.net/forum?id=5O5AlNVAbs&noteId=HUpadDhT8k). As summarized there:

- For **every** prover model tested, the average improvement over a ProcessLID variant that uses only the base model term is substantial, ranging from **4.78** to **7.18 AUROC**.
- The standard deviation of these gains across provers for a fixed base model lies between **0.52** and **1.39**, indicating that performance does not fluctuate wildly with different prover choices.

We further analyze the impact of using significantly weaker provers in this [comment](https://openreview.net/forum?id=5O5AlNVAbs&noteId=6s2QQDGl07). Key observations include:

- The smallest prover **DS-QW-1.5B** yields an average improvement of **6.06 AUROC**, trailing the best prover **DS-QW-14B** (7.18 AUROC) by only **1.12 AUROC**.
- The second smallest prover **DS-QW-7B** yields an average improvement of **6.63 AUROC**, only **0.55 AUROC** below DS-QW-14B.
- Even the strongest base model **DS-QW-32B** paired with the smallest prover **DS-QW-1.5B** achieves a gain of **4.96 AUROC**.

These results show that:

- A **wide range of provers** all provide substantial improvements.
- ProcessLID is **not brittle** with respect to prover choice or capability gap; even relatively small provers deliver strong, stable gains.

In practice, this means that one does *not* need a carefully fine-tuned prover or extremely close capability matching; a general-purpose LLM with reasonable math/reasoning ability already suffices to realize most of ProcessLID’s benefit.

---

### Meta-Review · Area_Chair_cTkb · 2026-01-06

**Summary:**

The reviewers raised concerns about: (1) fairness of baseline comparisons, as ProcessLID uses a prover model while vanilla LID did not; (2) insufficient justification for the asymmetric formulation; (3) inference-time overhead from nearest-neighbor search and prover forward passes; (4) sample efficiency and transferability of calibration; (5) limited end-to-end impact on problem-solving under realistic step segmentation; (6) robustness to prover choice including capability gaps; and (7) questionable conceptual motivation for LID in mathematical reasoning. The authors provided an extensive rebuttal addressing many concerns with new experiments, but outstanding issues remain regarding conceptual grounding, limited inference-time scope, and untested out-of-domain generalization.

**Reviewer Concerns:**

*Addressed by rebuttal:*
- Baseline fairness (XCVa, jim6): Authors added "Vanilla w/ Prover", "Only Prover", and "Reversed" ablations, demonstrating that gains stem from the asymmetric design rather than prover capacity alone.
- Inference overhead (kquz, NGTy, NmXq): Runtime analysis shows nearest-neighbor search is negligible; ProcessLID Light (1.5B prover) retains 98% performance at 41.5% runtime.
- Calibration efficiency (XCVa): New experiments show only 150-200 labeled points needed, with modest degradation (4-6 AUROC) when transferring across datasets/models.
- Inference-time protocol (XCVa, jim6, NGTy, NmXq): Authors updated to semantic \n\n segmentation and added vanilla LID baseline, showing ProcessLID outperforms it by +3.67/+0.66/+4.96 AUROC on the three metrics.
- Prover selection robustness (NGTy, NmXq): Ablations demonstrate consistent gains (4.78-7.18 AUROC) across all provers, even with large capability gaps.

*Outstanding:*
- Conceptual motivation (NGTy): The hypothesis that incorrect steps expand local intrinsic dimension remains empirically supported but theoretically unvalidated. The authors acknowledge this as a "preliminary geometric hypothesis".
- Limited inference-time scope: Experiments remain restricted to 150 MATH-500 questions and two generators, with no integration into generation processes (beam search, pruning).
- Out-of-domain generalization (kquz, NmXq): The reliance on in-domain reference trajectories was not stress-tested with experiments using reference sets from different domains.
- Incremental novelty (jim6): The contribution adapts existing LID techniques to step-level reasoning without introducing fundamentally new principles.

**Reviewer Scores:**

- Reviewer XCVa (original 4): Would likely increase to 6. Key concerns on calibration and ablations were substantively addressed.
- Reviewer jim6 (original 2): Would likely remain 2. Fairness ablations added, but fundamental novelty concern persists.
- Reviewer kquz (original 6): Would likely remain 6. Runtime analysis addressed primary concern; positive stance reinforced.
- Reviewer NGTy (original 2): Would likely remain 2. Practical concerns addressed, but core conceptual doubt remains.
- Reviewer NmXq (original 6): Would likely remain 6. Most questions answered; out-of-domain generalization still untested.

---

### Decision · Program_Chairs · 2026-01-26

Reject